# The major β-catenin/E-cadherin junctional binding site is a primary molecular mechano-transductor of differentiation *in vivo*

Jens-Christian Röper[1], Démosthène Mitrossilis[1†], Guillaume Stirnemann[2†], François Waharte[3], Isabel Brito[4], Maria-Elena Fernandez-Sanchez[1], Marc Baaden[2], Jean Salamero[3], Emmanuel Farge[1]*

[1]Mechanics and Genetics of Embryonic and Tumoral Development, Institut Curie, INSERM, CNRS UMR 168, PSL University, Paris, France; [2]CNRS Laboratoire de Biochimie Théorique, Institut de Biologie Physico-Chimique, PSL University, Paris, France; [3]Space-Time Imaging of Endomembranes Dynamics, Cell and Tissue Imaging Facility, Institut Curie, CNRS UMR 144, PSL University, Inria, France; [4]CBIO-Centre for Computational Biology, MINES ParisTech, Institut Curie, INSERM, PSL University, Paris, France

**Abstract** *In vivo*, the primary molecular mechanotransductive events mechanically initiating cell differentiation remain unknown. Here we find the molecular stretching of the highly conserved Y654-β-catenin-D665-E-cadherin binding site as mechanically induced by tissue strain. It triggers the increase of accessibility of the Y654 site, target of the Src42A kinase phosphorylation leading to irreversible unbinding. Molecular dynamics simulations of the β-catenin/E-cadherin complex under a force mimicking a 6 pN physiological mechanical strain predict a local 45% stretching between the two α-helices linked by the site and a 15% increase in accessibility of the phosphorylation site. Both are quantitatively observed using FRET lifetime imaging and non-phospho Y654 specific antibody labelling, in response to the mechanical strains developed by endogenous and magnetically mimicked early mesoderm invagination of gastrulating *Drosophila* embryos. This is followed by the predicted release of 16% of β-catenin from junctions, observed in FRAP, which initiates the mechanical activation of the β-catenin pathway process.
DOI: https://doi.org/10.7554/eLife.33381.001

*For correspondence:
efarge@curie.fr

†These authors contributed equally to this work

**Competing interests:** The authors declare that no competing interests exist.

## Introduction

Mechanical stresses are inherent to multi-cellular tissues in organisms, and can regulate signalling pathways involved both in their biochemical patterning and biomechanical morphogenesis (*Brouzés and Farge, 2004*; *Chan et al., 2017*; *Mammoto and Ingber, 2010*; *Wozniak and Chen, 2009*). Mechanical regulation is involved not only in physiological developmental processes, but also in pathological processes such as in tumorigenesis (*Mitrossilis et al., 2017*; *Dupont et al., 2011*; *Baeyens et al., 2015*; *Yang et al., 2016*; *Wu et al., 2015*; *Engler et al., 2006*; *Vogel and Sheetz, 2009*; *McBeath et al., 2004*; *Paszek et al., 2014*; *Uyttewaal et al., 2012*; *Zhang et al., 2011*; *Farge, 2003*). Such regulation is based on molecular mechanotransductional processes, which consist in the translation of mechanical strains submitted to living cells into biochemical signals (*Gospodarowicz et al., 1978*; *Nakache and Gaub, 1988*; *Kanno et al., 2007*; *Bissell et al., 2005*; *Rubashkin et al., 2014*; *Delarue et al., 2014*; *Liu et al., 2015*; *Mammoto et al., 2009*; *Yan et al., 2012*). The molecular basis of mechanotransduction is the activation or modulation of biochemical

reactions in response to cell or tissue mechanical strain-deformations. These are transmitted by the stress to subcellular mesoscopic structures or individual biomolecules. Mechanical induction of biochemical reactions is thus enabled by the coupling of the two major properties of molecular living matter: its high biochemical reactivity, and its high physical deformability. Since free-energy barriers between protein conformations are usually smaller than a few tens of kT (*Lannon et al., 2012*) (*i.e* several ten times the molecular thermal energy), mechanical stresses applied to biomolecular structures can *a priori* easily trigger conformational changes. Such deformations can subsequently lead to changes in the molecular biochemical reactivity and states, thereby initiating the activation or modulation of biochemical signal-transduction pathways (*Brujic et al., 2007*; *Fernandez-Sanchez et al., 2015a*; *Grashoff et al., 2010*; *Vargas et al., 2001*). Mechanical stresses have been shown to initiate a variety of molecular protein-conformation, or membrane shape changes, in cell culture. One example is the rendering of accessibility of the phosphorylation target site for kinases, like for Src-kinase, in the p130Cas protein involved in tumorigenic cell migration, in response to mechanical stress (*Sawada et al., 2006*). Another example is the mechanically induced inhibition of the formation of endocytic vesicles that lead to the inhibition or the degradation of interactions of secreted ligand-receptor complexes and thus activate or enhance downstream signalling pathways. This was shown in cell culture for BMP2 inducing myoblast/osteoblast trans-differentiation (*Rauch et al., 2002*).

However, none of the underlying molecular mechanism of mechanotransduction characterized so far has been found to be functionally involved in cell specification and differentiation, *in vivo*.

Among the many mechanosensitive pathways, the β-catenin (β-cat) pathway is mechanotransductively involved in major biochemical patterning processes in both developmental and tumorous contexts (*Fernandez-Sanchez et al., 2015a*; *Farge, 2003*; *Desprat et al., 2008*; *Brunet et al., 2013*; *Benham-Pyle et al., 2015*; *Fernández-Sánchez et al., 2015b*; *Samuel et al., 2011*; *Whitehead et al., 2008*; *Mouw et al., 2014*). Initially, it was found that mechanical activation of β-cat and downstream transcriptional signaling is vitally involved in embryonic anterior endoderm specification in response to the first morphogenetic movements of gastrulation (*Farge, 2003*; *Desprat et al., 2008*) In addition, it is involved in *ex-vivo* reconstructed epithelial cells in response to mechanical stretching (*Benham-Pyle et al., 2015*; *Benham-Pyle et al., 2016*), in tumor progression in healthy tissues in response to neighboring tumor growth pressure and stiffness (*Fernández-Sánchez et al., 2015b*; *Samuel et al., 2011*; *Whitehead et al., 2008*), or in follicle development initiation (*Shyer et al., 2017*). In both zebrafish and *Drosophila* embryos, early mesoderm differentiation was also found to be mechanically induced by the very first morphogenetic movements of embryogenesis. This induction occurs through the mechanical activation of the β-cat pathway, where the mechanical strain generated by the onset of embryonic morphogenesis mechanically induces the Src-family kinase (SFK) dependent phosphorylation of the highly conserved β-cat Y654 site (*Brunet et al., 2013*; *Fernández-Sánchez et al., 2015b*), a major site of interaction with E-cadherin (E-cad) in adherens junctions (AJ) (*Roura et al., 1999*; *van Veelen et al., 2011*). This chemical modification leads to β-cat release from AJ and its translocation to the nucleus, where it activates the transcription of early mesodermal key specification genes (*Brunet et al., 2013*). The molecular mechanism of mechanotransduction that mechanically initiates such SFK-dependent β-cat phosphorylation *in vivo*, as well as its primary mechano-sensor, are still unknown.

Here we identified the primary mechanotransductive molecular process and the mechano-sensor. We tested *in vivo* the hypothesis that the major Y654-β-cat/E-cad-D665 interaction site of the AJ complex (*Fernandez-Sanchez et al., 2015a*) could be, by itself, the primary mechanosensor that activates the β-cat pathway through its mechanical opening. This hypothesis implied to characterize the mechanical opening of the molecular interaction between the Y654 β-cat tyrosine with the D665-E-cad aspartic acid. This mechanical event would favour phosphorylation by Src42A, that is known to lead to a 86% decrease in the Y654 β-cat-D665-E-cadherin affinity (*Roura et al., 1999*; *van Veelen et al., 2011*). It further releases β-cat into the cytosol and nucleus, where it targets functional patterning gene transcription during mesoderm invagination at gastrulation (*Brunet et al., 2013*).

## Results

β-cat is highly conserved across metazoan, with a central armadillo repeat domain consisting of twelve repeats (*Figure 1a*). These repeats form a spring like superhelix featuring a long, positively charged groove that interacts with several key residues of the cytoplasmic tail of E-cad (*Figure 1—*

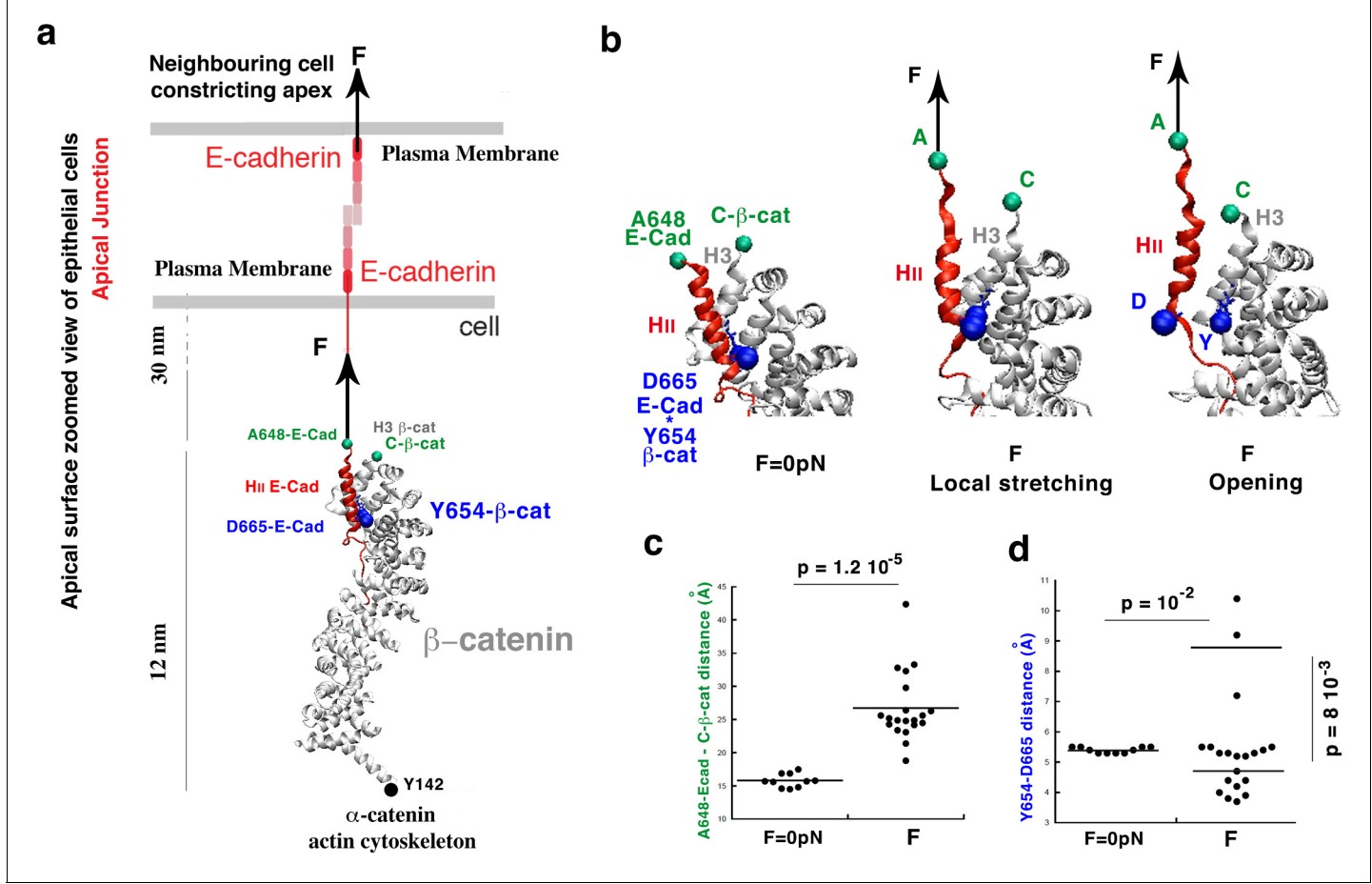

**Figure 1.** Molecular dynamics simulations of the β-cat/E-cad complex under mechanical strain. (**a**) Stretching of the apical junctional β-cat/E-cad complex in response to an external force applied from the outside of the cell by constricting apexes of neighbouring cells. (**b**) Zoom on the local stretching between HII-E-cad and H3−β-cat helices, linked by the Y654-β-cat-D665-E-cad interacting site, leading to its opening. (**c**) Quantification of the HII-E-cad-HII-E-cad local stretching mean distance between A648-E-cad and C-β-cat under a force F equivalent to 6 pN strain compared to control. (**d**) Quantification of the mean distance between Y654-β-cat and D665-E-cad (distance between the OH of the Y654 phenol and the carbon of COO- of the D665 aspartate). Controls (F = 0 pN): N = 10. Strained: N = 20. Statistical test: Mann-Whitney.

DOI: https://doi.org/10.7554/eLife.33381.002

The following source data and figure supplements are available for figure 1:

**Source data 1.** Y654-ß-cat-D665-E-cad interacting site distances under local H3-ß-cat-HII-E-cad stretching due to the global stretching of the ß-cat-E-cad complex, molecular dynamics simulation.

DOI: https://doi.org/10.7554/eLife.33381.006

**Figure supplement 1.** Molecular basis of FRET/FLIM design.

DOI: https://doi.org/10.7554/eLife.33381.003

**Figure supplement 2.** Y654-ß-cat-D665-E-cad interacting site distances under local H3-ß-cat-HII-E-cad stretching due to the global stretching of the ß-cat-E-cad complex.

DOI: https://doi.org/10.7554/eLife.33381.004

**Figure supplement 3.** Conservation of both the Y654-β-cat and D665-E-cad sites across metazoans.

DOI: https://doi.org/10.7554/eLife.33381.005

*figure supplement 1a*) (*Huber and Weis, 2001*; *Orsulic and Peifer, 1996*). The strength of interaction is dynamically modulated, mostly by the phosphorylation of β-cat on tyrosine Y654 (*Figure 1— figure supplement 1b,c* in red), reducing the affinity of E-cad to β-cat by approximately 86% (*Roura et al., 1999*; *van Veelen et al., 2011*). In mammalian cells, phosphorylation of β-cat Y654 has been shown to involve SFK (*Roura et al., 1999*). In *Drosophila*, the Y667 site of the β-cat homologue Armadillo shows sequence similarity to the conserved Src-phosphorylation site at Y654 in

mouse (**Brunet et al., 2013**; **Roura et al., 1999**; **van Veelen et al., 2011**) (**Figure 1—figure supplement 1d**). Its *Drosophila* E-cad D1170 interacting site similarly shows sequence conservation with the D665 mice site (**Figure 1—figure supplement 1e**) (**Huber and Weis, 2001**).

To test the hypothesis of a mechanical opening of the Y654-β-cat/E-cad-D665 site as the primary mechantransductive molecular event initiating the mechanical activation of the β-cat pathway, and gain insight at the molecular level, we performed molecular dynamics (MD) simulations of the β-cat-E-cad complex submitted to a stretching uniaxial force (**Huang and MacKerell, 2013**; **Phillips et al., 2005**).

The force is applied on E-Cad by both the cell's and the neighbouring cell's apical constrictions that drive *Drosophila* embryos mesoderm invagination (**Sweeton et al., 1991**). It thus acts perpendicular to the plasma membrane directed to the outside, with β-cat fixed at its Y142 interacting site with α-catenin which is connected to the Actin cytoskeleton (**Xu and Kimelman, 2007**) (**Figure 1a**). Note that the timescales of conventional MD simulations are currently limited to the hundreds of nanoseconds to the microsecond. To observe protein unfolding, or conformational changes in response to mechanical force on such a reduced timescale, it is therefore common to use forces that are higher than experimental ones. Due to this high force, the same events that biologically occur on a timescale that is not accessible by the MD simulations are sped up and hence become amenable to molecular simulations.

Here, we generated 20 independent simulations for a total simulation time of 30 ns each (600 ns total) on a protein complex of significant size (more than 500 residues). Our simulations under strain employed a constant force of 150 pN in order to observe relevant conformational changes on the 30-ns timescale of the simulation (**Figure 1—figure supplement 2a**). We estimated *a posteriori* the biological force that is mimicked by our procedure as follows. The mean elongation of the simulated complex showing the opening of the Y654-D665 site on the timescale of 12 ns is of 6 nm (**Figure 1—figure supplement 2a**). The proteins including β-cat are formed by α-helices only, with a modulus of elasticity of $K_h$ of an α-helix on the order of 1 pN/nm (**Kim et al., 2010**). Therefore we find that this elongation corresponds to a force of 6 pN, consistent with the 2-3 pN characteristic force of E-cad in adherent cells without external force (**Borghi et al., 2012**).

After a global extension of 8 nm of the overall complex, a local and fine stretching deformation is observed between the A648 site of the α-helix HII-E-cad and the C-terminus of the α-helix H3-β-cat, which are linked by the Y654-β-cat-D665-E-cad (Y654-D665) interacting site, of 1.25 ± 0.6 nm (45% strain). This change led, at 6 nm of extension mimicking an actual strain of 6 pN, to a discrete opening of the Y654-D665 interacting site of 0.42 ± 0.08 nm in 3 simulations out of 20, namely in 15% of cases (**Figure 1b,c,d**, **Videos 1** and **2** and **Figure 1—figure supplement 2a,b,c, d**, respectively; See **Figure 1c,d** and **Figure 1— source data 1**).

We then tested *in vivo* the prediction of the existence of a strain deformation in the β-cat-E-cad complex of 1.25 nm between the A648-E-cad and the β-cat-C-terminus, due to mesoderm invagination in *Drosophila* embryos. In this case, mechanical induction of the Y654-β-cat

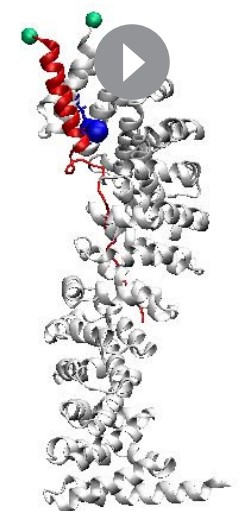

**Video 1.** Example of one MD simulation of the β-cat/ E-cad complex with no external force applied (30 ns).
DOI: https://doi.org/10.7554/eLife.33381.007

phosphorylation by mesoderm invagination is specifically required for mesoderm specification (**Brunet et al., 2013**). To do so, we carried out fluorescence lifetime imaging experiments (FLIM) to measure fluorescence resonance energy transfer (FRET) efficiency between the loop domain of E-cad region II directly linked to A648 and β-cat-C-terminus. For the purpose of introducing fluorescent groups, we used embryos that express a version of β-cat tagged with GFP as a donor at the C-terminus. The acceptor is an injected antibody (JCAb20) fluorescently labelled with Alexa555 and specifically designed against the linear domain of region II in the E-cadherin protein, which is directly linked to A648-E-cad (acceptor) (**Huber and Weis, 2001**) (**Figure 1—figure supplement 1**, **Figure 2a** and **Figure 2—figure supplement 1**. Schematized in **Figure 2a**, the two fluorochromes are linked to the two green spheres of **Figure 1** simulation, see Materials and methods).

The choice of the E-cad site was based on the crystalline structure of the β-cat/E-cad complex of mice (**Huber and Weis, 2001**), and on the high conservation of the sequence of E-cad around D665 between *Drosophila* and mice (**Figure 1—figure supplement 1e**). The JCAb20 antibody targets an E-cad loop at 1143–1158 (*in Drosophila*) neighbouring directly the alpha helix of region II. The latter contains the D665 (D1170 in *Drosophila*), which interacts with the Y654 site of β-cat (Y667 in *Drosophila*), and is directly labelled with Alexa 555 (in orange in **Figure 1—figure supplement 1c** and **Figure 1—figure supplement 1a**, from Protein Data Bank and **Huber and Weis, 2001**, with accession code 1I7X, schematized in red **Figure 2a**) (**Huber and Weis, 2001**). The free E-cad 1143–1158 loop of the crystalline structure is shown in purple and green in **Figure 1—figure supplement 1a**. The antibody was tested for its specificity by immunostaining, western-blot analysis and immunoprecipitation (see **Figure 2—figure supplement 1a, b**). The peptide sequence of the immunogen was checked for possible similarities with other proteins by a standard protein-protein BLAST search against the *Drosophila* proteome (taxid: 7227) (see **Table 1**). The GFP, about 4 nm long, is located at the C-terminus of the β-cat (schematized in green in **Figure 2a**) (**Peifer, 2004**).

At the time of antibody injection, the cells in the embryo still form a syncytium, that is they are open to the same internal yolk volume, which allows the antibody to diffuse into any cell. Afterwards, cells are closed in a process called cellularization followed by the first embryonic morphogenetic movement, the invagination of the mesodermal tissue during gastrulation (**Wieschaus and Nusslein-Volhard, 1998**). FRET experiments were performed by measuring the lifetime of the β-cat-GFP donor (FLIM), as the E-cad Alexa555 labelled antibody acceptor acts as a quencher that reduces the lifetime of the donor. A relatively low lifetime indicates higher FRET efficiency and closer distances, while a lifetime close to the normal lifetime of the fluorophore indicates low FRET efficiency and higher distances (**Figure 2a**).

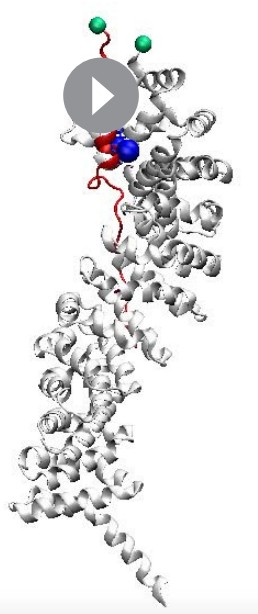

**Video 2.** Example of one MD simulation of the β-cat/E-cad complex with a force equivalent to an actual 6 pN external strain applied at the A648 site of the E-cad in the direction of the plasma membrane in 30 ns.
DOI: https://doi.org/10.7554/eLife.33381.008

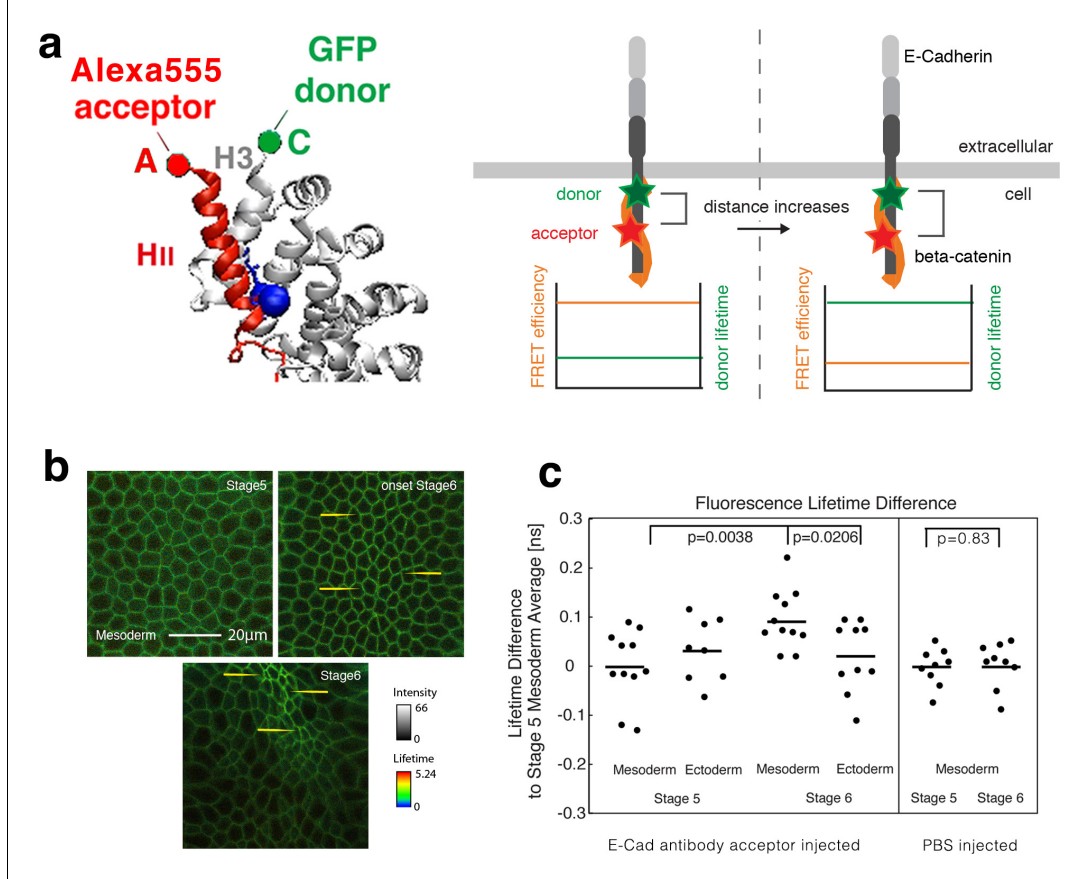

**Figure 2.** Fluorescence lifetime measurements of the β-cat-C-terminus-GFP donor in the presence of the Alexa555 antibody against E-cad linear region II as acceptor mesoderm invagination. (a) Illustration of FRET experiments performed at two different stages of development when the strain in the tissue builds up. Embryos expressing β-cat-GFP (donor) are injected with anti-E-cad antibodies directly labelled with Alexa555 (acceptor) for FRET experiments. FRET experiments are performed by analysing the lifetime of the donor. Images showing the lifetime of GFP donor: the FRET efficiency will decrease when the two sites that are attached to the fluorophore, become more separated by increased strain in the molecule. Structure based on reference (*Huber and Weis, 2001*). (b) Invaginating embryo showing the invaginated furrow at stage 6 with a longer junctional lifetime compared to the surrounding ectodermal tissue and to stage 5 mesoderm cells (examples shown with yellow arrows). (c) Left- Quantification of the lifetime difference at different stages and different tissues. ANOVA test p value: p=0.0156. T-test p-values are of p=0.021 for mesoderm (N = 11) versus ectoderm (N = 10) at stage 6 and 0.004 for mesoderm at stage 6 versus stage 5 (N = 11). No significant change was observed in the ectoderm at stage 6 (N = 10) compared to stage 5 (N = 8) (p=0.74), and between the mesoderm and the ectoderm at stage 5 (p=0.33). In some embryos, the ectoderm could not be imaged properly. Right- Differences of lifetime of β-cat-GFP expressing Drosophila embryos injected with PBS without antibody in the mesoderm at stage 5 (N = 9) and stage 6 (N = 9) respectively (p=0.83). As expected, PBS injection without the antibody acceptor lead to a lifetime of 2.600 ± 0.039 ns, higher compared to 2.473 ± 0.074 ns with the antibody at stage 5 (see *Figure 2c*-source data2, p=0.00016). Statistical tests are t-student.
DOI: https://doi.org/10.7554/eLife.33381.009

The following source data and figure supplements are available for figure 2:

**Source data 1.** Fluorescence lifetime measurements of the β-cat-C-terminus-GFP donor in the presence of the E-cad linear region II Alexa555 acceptor during mesoderm invagination.
DOI: https://doi.org/10.7554/eLife.33381.012

**Source data 2.** FLIM signal of the β-cat-GFP donor in the absence of acceptor during mesoderm invagination.
DOI: https://doi.org/10.7554/eLife.33381.013

**Figure supplement 1.** Specific labeling of junctional E-cad with JCAb20.
DOI: https://doi.org/10.7554/eLife.33381.010

**Figure supplement 2.** FLIM signal of the β-cat-GFP donor in the absence of acceptor during mesoderm invagination.
DOI: https://doi.org/10.7554/eLife.33381.011

**Table 1.** List of hits resulting from a standard protein-protein BLAST search against the *Drosophila* proteome (taxid: 7227) for the immunogen peptide sequence used to generate the JCAb20 antibody.

| Description | Max score | Total score | Query cover | E value | Ident | Accession |
|---|---|---|---|---|---|---|
| DE-cadherin [Drosophila melanogaster] | 53.2 | 66.6 | 93% | 1.00E-11 | 100% | BAA05942.1 |
| Shotgun [Drosophila melanogaster] | 53.2 | 66.6 | 93% | 1.00E-11 | 100% | NP_476722.1 |
| GM13889p [Drosophila melanogaster] | 24.8 | 24.8 | 56% | 0.13 | 50% | AAM50217.1 |
| GH21511p [Drosophila melanogaster] | 24.8 | 24.8 | 56% | 0.14 | 67% | AAK92991.1 |
| Eukaryotic translation initiation factor 3 p66 subunit [Drosophila melanogaster] | 24.8 | 24.8 | 56% | 0.14 | 50% | AAF37264.1 |
| Eukaryotic initiation factor 3 p66 subunit [Drosophila melanogaster] | 24.8 | 24.8 | 56% | 0.14 | 50% | NP_524463.2 |
| Extended synaptotagmin-like protein 2 ortholog, isoform A [Drosophila melanogaster] | 24.8 | 42.4 | 62% | 0.14 | 67% | NP_733010.1 |
| Extended synaptotagmin-like protein 2 ortholog, isoform D [Drosophila melanogaster] | 24.8 | 42.4 | 62% | 0.14 | 67% | NP_001262921.1 |
| Extended synaptotagmin-like protein 2 ortholog, isoform B [Drosophila melanogaster] | 24.8 | 42.4 | 62% | 0.14 | 67% | NP_733011.2 |
| RE26910p [Drosophila melanogaster] | 24.8 | 42.4 | 62% | 0.14 | 67% | ABY21755.1 |
| Extended synaptotagmin-like protein 2 ortholog, isoform C [Drosophila melanogaster] | 24.8 | 42.4 | 62% | 0.14 | 67% | NP_001247287.1 |
| Uncharacterized protein Dmel_CG11920 [Drosophila melanogaster] | 23.5 | 23.5 | 62% | 0.38 | 70% | NP_651335.2 |
| GM02712p [Drosophila melanogaster] | 23.5 | 23.5 | 62% | 0.38 | 70% | AAL28396.1 |
| Mind bomb 2, isoform A [Drosophila melanogaster] | 23.5 | 36.9 | 93% | 0.39 | 70% | NP_609933.1 |
| Scribbled [Drosophila melanogaster] | 23.5 | 23.5 | 37% | 0.39 | 100% | AAO32791.1 |
| Vartul-2 protein [Drosophila melanogaster] | 23.5 | 40.3 | 37% | 0.39 | 100% | CAB71137.1 |
| Scribbled, isoform C [Drosophila melanogaster] | 23.5 | 40.3 | 37% | 0.39 | 100% | NP_733156.1 |
| Scribbled, isoform I [Drosophila melanogaster] | 23.5 | 40.3 | 37% | 0.39 | 100% | NP_001014669.1 |
| Scribbled, isoform P [Drosophila melanogaster] | 23.5 | 40.3 | 37% | 0.39 | 100% | NP_733155.2 |
| Scribble [Drosophila melanogaster] | 23.5 | 54.5 | 75% | 0.39 | 100% | AAF26357.2 |
| Scribbled, isoform A [Drosophila melanogaster] | 23.5 | 40.3 | 37% | 0.39 | 100% | NP_733154.1 |
| RE02389p [Drosophila melanogaster] | 23.5 | 40.3 | 37% | 0.39 | 100% | AAT94469.1 |
| Vartul-1 protein [Drosophila melanogaster] | 23.5 | 40.3 | 37% | 0.39 | 100% | CAB70601.1 |
| Scribbled, isoform U [Drosophila melanogaster] | 23.5 | 40.3 | 37% | 0.39 | 100% | NP_001262989.1 |
| Scribbled [Drosophila melanogaster] | 23.5 | 40.3 | 37% | 0.39 | 100% | AAO32792.1 |

*Table 1 continued on next page*

*Table 1 continued*

| Description | Max score | Total score | Query cover | E value | Ident | Accession |
|---|---|---|---|---|---|---|
| Scribbled, isoform D [Drosophila melanogaster] | 23.5 | 40.3 | 37% | 0.39 | 100% | NP_524754.2 |
| Scribbled, isoform H [Drosophila melanogaster] | 23.5 | 40.3 | 37% | 0.39 | 100% | NP_001014670.2 |
| Scribbled, isoform R [Drosophila melanogaster] | 23.5 | 40.3 | 37% | 0.39 | 100% | NP_001247322.1 |
| Scribbled, isoform K [Drosophila melanogaster] | 23.5 | 40.3 | 37% | 0.39 | 100% | NP_001247318.1 |
| Scribbled, isoform J [Drosophila melanogaster] | 23.5 | 40.3 | 37% | 0.39 | 100% | NP_001163745.1 |
| Scribbled, isoform T [Drosophila melanogaster] | 23.5 | 40.3 | 37% | 0.39 | 100% | NP_001262988.1 |
| Scribbled, isoform M [Drosophila melanogaster] | 23.5 | 40.3 | 37% | 0.39 | 100% | NP_001163747.1 |
| Scribbled, isoform O [Drosophila melanogaster] | 23.5 | 40.3 | 37% | 0.39 | 100% | NP_001247320.1 |
| Scribbled, isoform N [Drosophila melanogaster] | 23.5 | 40.3 | 37% | 0.39 | 100% | NP_001247319.1 |
| Scribbled, isoform Q [Drosophila melanogaster] | 23.5 | 40.3 | 37% | 0.39 | 100% | NP_001247321.1 |
| Scribbled, isoform L [Drosophila melanogaster] | 23.5 | 40.3 | 37% | 0.39 | 100% | NP_001163746.1 |
| TPA: HDC02662 [Drosophila melanogaster] | 23.1 | 23.1 | 56% | 0.54 | 62% | DAA03664.1 |
| Uncharacterized protein Dmel_CG10103, isoform D [Drosophila melanogaster] | 23.1 | 35.6 | 43% | 0.55 | 86% | NP_001261485.1 |
| Uncharacterized protein Dmel_CG10103, isoform E [Drosophila melanogaster] | 23.1 | 35.6 | 43% | 0.55 | 86% | NP_001261486.1 |
| Uncharacterized protein Dmel_CG10103, isoform C [Drosophila melanogaster] | 23.1 | 35.6 | 43% | 0.55 | 86% | NP_001261484.1 |
| Uncharacterized protein Dmel_CG10103, isoform B [Drosophila melanogaster] | 23.1 | 35.6 | 43% | 0.55 | 86% | NP_001261483.1 |
| Uncharacterized protein Dmel_CG10103, isoform A [Drosophila melanogaster] | 23.1 | 35.6 | 43% | 0.55 | 86% | NP_648058.1 |
| Uncharacterized protein Dmel_CG11883, isoform B [Drosophila melanogaster] | 22.7 | 40.7 | 43% | 0.78 | 86% | NP_610598.1 |
| Uncharacterized protein Dmel_CG11883, isoform A [Drosophila melanogaster] | 22.7 | 40.7 | 43% | 0.78 | 86% | NP_724960.1 |
| Uncharacterized protein Dmel_CG11883, isoform C [Drosophila melanogaster] | 22.7 | 40.7 | 43% | 0.78 | 86% | NP_001097260.1 |
| Peptidylprolyl cis-trans isomerase [Drosophila melanogaster] | 22.7 | 22.7 | 50% | 0.78 | 63% | AAN39118.1 |
| Moca-cyp, isoform A [Drosophila melanogaster] | 22.7 | 22.7 | 50% | 0.78 | 63% | NP_733246.1 |
| RE23622p [Drosophila melanogaster] | 22.7 | 22.7 | 50% | 0.78 | 63% | AAN71333.1 |
| GH13610p [Drosophila melanogaster] | 22.7 | 22.7 | 68% | 0.78 | 64% | ACV31093.1 |
| GH14650p [Drosophila melanogaster] | 22.7 | 22.7 | 68% | 0.78 | 64% | AAK92914.1 |
| SP2523 [Drosophila melanogaster] | 22.7 | 22.7 | 68% | 0.78 | 64% | AAF63503.1 |
| Ankyrin 2, isoform L [Drosophila melanogaster] | 22.7 | 22.7 | 68% | 0.78 | 64% | NP_729285.3 |

*Table 1 continued on next page*

*Table 1 continued*

| Description | Max score | Total score | Query cover | E value | Ident | Accession |
|---|---|---|---|---|---|---|
| Ankyrin 2, isoform F [Drosophila melanogaster] | 22.7 | 22.7 | 68% | 0.78 | 64% | NP_001097535.1 |
| Ankyrin 2, isoform J [Drosophila melanogaster] | 22.7 | 38.2 | 93% | 0.78 | 64% | NP_001097538.1 |
| Ankyrin 2, isoform T [Drosophila melanogaster] | 22.7 | 22.7 | 68% | 0.78 | 64% | NP_001189068.1 |
| Ankyrin 2, isoform P [Drosophila melanogaster] | 22.7 | 22.7 | 68% | 0.78 | 64% | NP_001189069.1 |
| Ankyrin 2, isoform Z [Drosophila melanogaster] | 22.7 | 22.7 | 68% | 0.78 | 64% | NP_001286971.1 |
| Ankyrin 2, isoform K [Drosophila melanogaster] | 22.7 | 22.7 | 68% | 0.78 | 64% | NP_001097539.1 |
| Ankyrin 2, isoform S [Drosophila melanogaster] | 22.7 | 38.2 | 93% | 0.78 | 64% | NP_001189064.1 |
| Ankyrin 2, isoform Q [Drosophila melanogaster] | 22.7 | 38.2 | 93% | 0.78 | 64% | NP_001189067.1 |
| Ankyrin 2, isoform V [Drosophila melanogaster] | 22.7 | 22.7 | 68% | 0.78 | 64% | NP_001261535.1 |
| Ankyrin 2, isoform R [Drosophila melanogaster] | 22.7 | 38.2 | 93% | 0.78 | 64% | NP_001189065.1 |
| Ankyrin 2, isoform U [Drosophila melanogaster] | 22.7 | 52.4 | 93% | 0.78 | 64% | NP_001189070.1 |
| Mei-P26 [Drosophila melanogaster] | 22.3 | 22.3 | 43% | 1.1 | 71% | AGL81505.1 |
| Mei-P26 [Drosophila melanogaster] | 22.3 | 22.3 | 43% | 1.1 | 71% | AGL81497.1 |
| Mei-P26 [Drosophila melanogaster] | 22.3 | 22.3 | 43% | 1.1 | 71% | AGL81501.1 |
| Mei-P26 [Drosophila melanogaster] | 22.3 | 22.3 | 43% | 1.1 | 71% | AGL81499.1 |
| Mei-P26 [Drosophila melanogaster] | 22.3 | 22.3 | 43% | 1.1 | 71% | AGL81498.1 |
| HL03718p [Drosophila melanogaster] | 22.3 | 22.3 | 37% | 1.1 | 100% | AAR82799.1 |
| BcDNA.GH10646 [Drosophila melanogaster] | 22.3 | 22.3 | 43% | 1.1 | 71% | AAD38636.1 |
| Meiotic P26, isoform A [Drosophila melanogaster] | 22.3 | 22.3 | 43% | 1.1 | 71% | NP_652022.1 |
| Pangolin, isoform J [Drosophila melanogaster] | 22.3 | 35.6 | 37% | 1.1 | 100% | NP_726528.2 |
| Meiotic P26, isoform C [Drosophila melanogaster] | 22.3 | 22.3 | 43% | 1.1 | 71% | NP_001259377.1 |
| LD23595p [Drosophila melanogaster] | 21.8 | 35.2 | 75% | 1.6 | 86% | AAM52675.1 |
| Updo [Drosophila melanogaster] | 21.8 | 21.8 | 62% | 1.6 | 50% | NP_610501.1 |
| SD19419p [Drosophila melanogaster] | 21.8 | 21.8 | 62% | 1.6 | 50% | AAM51098.1 |
| Uncharacterized protein Dmel_CG12093 [Drosophila melanogaster] | 21.8 | 35.2 | 75% | 1.6 | 86% | NP_647747.1 |
| CG3803 [Drosophila melanogaster] | 21.8 | 21.8 | 75% | 1.6 | 40% | NP_611855.1 |
| CG9837, isoform D [Drosophila melanogaster] | 21.4 | 21.4 | 43% | 2.2 | 71% | NP_001163557.2 |

*Table 1 continued on next page*

*Table 1 continued*

| Description | Max score | Total score | Query cover | E value | Ident | Accession |
|---|---|---|---|---|---|---|
| CG9837, isoform E [Drosophila melanogaster] | 21.4 | 38.6 | 56% | 2.2 | 71% | NP_649838.5 |
| Missing oocyte [Drosophila melanogaster] | 21.4 | 33.9 | 68% | 2.2 | 100% | NP_608656.1 |
| Uncharacterized protein Dmel_CG17258, isoform C [Drosophila melanogaster] | 21.4 | 21.4 | 62% | 2.2 | 58% | NP_001259980.1 |
| Uncharacterized protein Dmel_CG17258, isoform D [Drosophila melanogaster] | 21.4 | 21.4 | 62% | 2.2 | 58% | NP_001259981.1 |
| AT19777p [Drosophila melanogaster] | 21.4 | 21.4 | 62% | 2.2 | 58% | ABB36463.1 |
| Uncharacterized protein Dmel_CG17258, isoform B [Drosophila melanogaster] | 21.4 | 21.4 | 62% | 2.2 | 58% | NP_608742.2 |
| DUNC79 [Drosophila melanogaster] | 21.4 | 63.2 | 93% | 2.2 | 75% | ABI95804.1 |
| Unc79, isoform B [Drosophila melanogaster] | 21.4 | 63.2 | 93% | 2.2 | 75% | NP_001163652.1 |
| Unc79, isoform F [Drosophila melanogaster] | 21.4 | 63.2 | 93% | 2.2 | 75% | NP_001287404.1 |
| Unc79, isoform E [Drosophila melanogaster] | 21.4 | 63.2 | 93% | 2.2 | 75% | NP_001287403.1 |
| Unc79, isoform C [Drosophila melanogaster] | 21.4 | 63.2 | 93% | 2.2 | 75% | NP_650795.2 |
| CG9034, isoform A [Drosophila melanogaster] | 21 | 21 | 37% | 3 | 83% | NP_652538.1 |
| Uncharacterized protein Dmel_CG33796 [Drosophila melanogaster] | 21 | 21 | 43% | 3.1 | 71% | NP_001027128.1 |
| Nubbin [Drosophila melanogaster] | 21 | 39 | 68% | 3.1 | 64% | ABD33837.1 |
| IP05211p [Drosophila melanogaster] | 21 | 21 | 56% | 3.1 | 67% | ABC86455.1 |
| Uncharacterized protein Dmel_CG32373, isoform A [Drosophila melanogaster] | 21 | 21 | 37% | 3.1 | 83% | NP_729293.1 |
| RT09995p [Drosophila melanogaster] | 21 | 21 | 56% | 3.1 | 67% | ADN05223.1 |
| Glycogen binding subunit 76A, isoform A [Drosophila melanogaster] | 21 | 21 | 50% | 3.1 | 75% | NP_649104.1 |
| Veloren, isoform B [Drosophila melanogaster] | 21 | 21 | 68% | 3.1 | 55% | NP_788470.1 |
| Nubbin, isoform B [Drosophila melanogaster] | 21 | 39 | 68% | 3.1 | 64% | NP_001097153.1 |
| Shal interactor of Di-Leucine motif [Drosophila melanogaster] | 21 | 21 | 37% | 3.1 | 83% | NP_650431.1 |

DOI: https://doi.org/10.7554/eLife.33381.014

FLIM measurements were performed during the two following different developmental stages during gastrulation: just before mesoderm invagination (last 10 min of cellularization, late embryonic stage 5), and during invagination (stage 6) (*Figure 2b*) when the acto-myosin contractile mechanical forces acting along the AJ increase in the mesoderm (*Martin et al., 2010*). At stage 6, measurements showed an increase of the GFP lifetime in the mesoderm, observed by an increase of bright green and of yellow spots in junctions of the invaginating mesoderm (e.g. yellow arrows in *Figure 2b*), of 0.073 ± 0.029 ns compared to the ectoderm, and of 0.094 ± 0.059 ns compared to stage 5 mesoderm (2.473 ns) (*Figure 2c*, *Figure 2—source data 1*).

As a control, no change in the GFP donor lifetime was observed during gastrulation in the absence of injected antibody acceptor (E-cad Alexa 555 labelled AB), as well as in embryos injected with PBS alone in the absence of the antibody acceptor (*Figure 2—figure supplement 2a,b* and *Figure 2c*-right). Such a fine increase in lifetime is quantitatively in line with a mean increase in

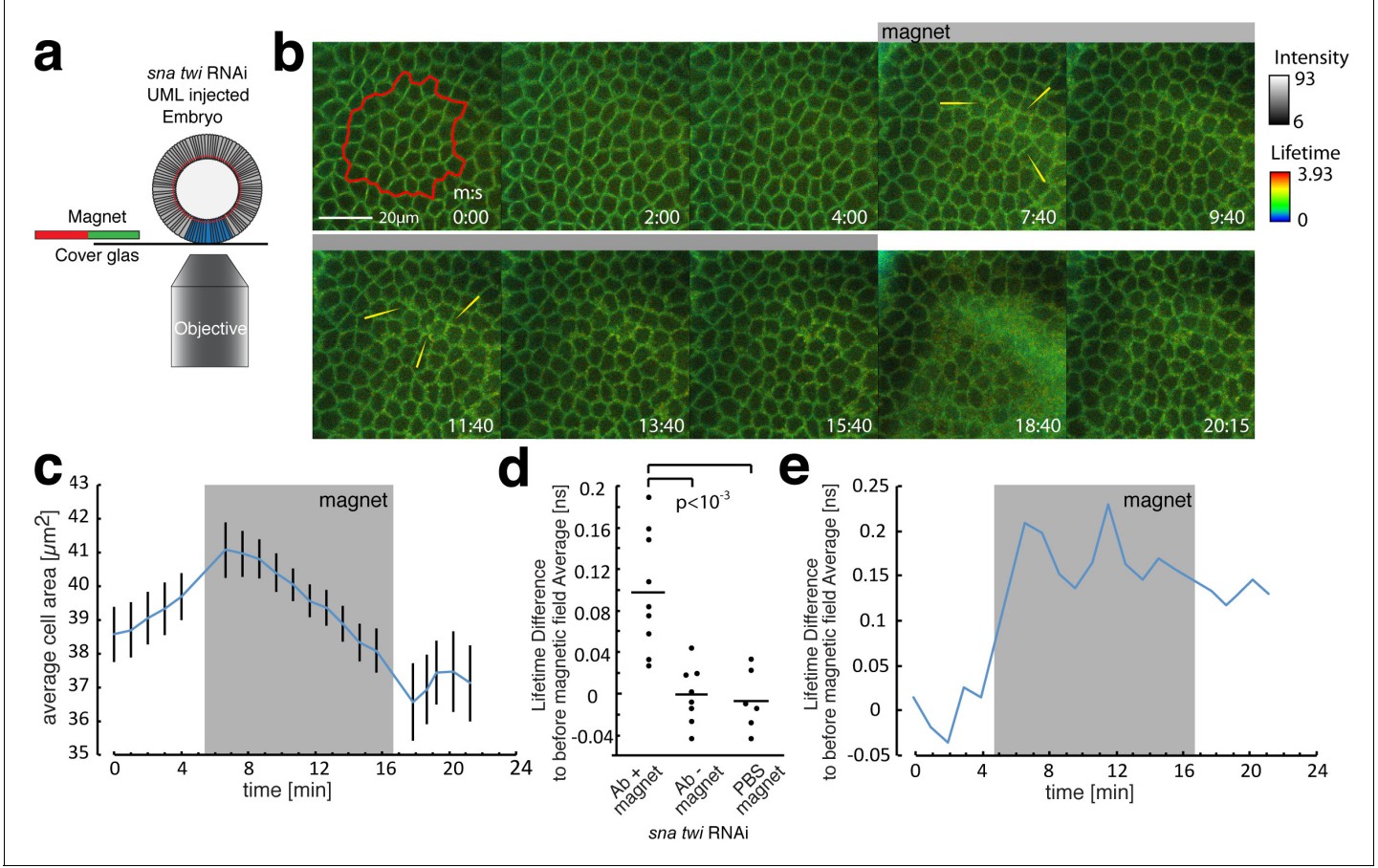

**Figure 3.** FLIM measurements in the mesoderm in response to magnetically induced cell apex constriction mimicking the onset of mesoderm invagination. (a) Illustration of the magnetic stimulation setup. The blue region corresponds to the mesoderm, in which UMLs where injected. (b) Time series of a magnetic stimulation experiment with β-cat-GFP expressing flies injected with siRNA for *snail* and *twist* as well as UMLs and Alexa555 labelled antibody against E-cad. The grey bar indicates when magnetic force was applied. Yellow arrows indicate examples of lifetime increase in the junctions. Times are indicated in min/s. (c) Quantification of the cell apex area for the experiment shown in b. Note that the area is decreasing in the *sna twi* RNAi background when the magnetic field is applied. (d) Lifetime difference of stage 6 to the average lifetime before magnetic field application of stage 5, of N = 9 experiments with magnet applied and N = 8 without magnet applied as well as N = 6 experiments with PBS instead of antibody injected and magnet applied. Whereas the experiments without magnetic field applied, or PBS instead of antibody injected, show no difference, the one with antibody and magnetic field applied show a significant increase of the lifetime, p value = 0.00061 (t-students test). The absence of increase of the lifetime difference within *sna twi* RNAi non-invaginating conditions without magnet between stage 5 and stage 6 is characterized by the p-value p=0.0003 compared to the increase between stage 5 and 6 in the invaginating WT of *Figure 2c*-left, and of p=0.75 compared to the lifetime difference in the invaginating WT injected with PBS of *Figure 2c*-right (t-students test). (e) Lifetime difference to the average lifetime before the magnetic field was applied for the experiment shown in. Grey box indicates when the magnetic field was applied.

DOI: https://doi.org/10.7554/eLife.33381.015

The following source data is available for figure 3:

**Source data 1.** FLIM measurements in the mesoderm in response to magnetically induced cell apex constriction mimicking the onset of mesoderm invagination.

DOI: https://doi.org/10.7554/eLife.33381.016

distance on the order of ~1 nm (see 'Calculation of the strain from FRET efficiency variations' in the Materials and methods section) between the GFP tagged β-cat and the fluorescently tagged E-cad antibody. Indeed, such ~1 nm stretching is quantitatively predicted by the MD simulations (*Figure 1c*). The lifetime increase observed specifically in the mesoderm at gastrulation thus shows the existence of a mechanical stretching strain between the fluorescently labelled linear domain II of E-cad, and the C-terminus of β-cat, respectively, in the invaginating mesoderm.

Note that the lifetime of the controls with GFP alone in the embryo junctions and with PBS injection without antibody, of 2.60 ± 0.04 ns, (*Figure 2—figure supplement 2a,b*, *Figure 2—source data 2* and *Figure 2—source data 1*), was consistently higher than in the presence of the acceptor before stretching, of 2.473 ± 0.074 ns (*Figure 2—source data 1*). These values are similar to the lifetime of GFP tagged proteins measured *in cellula*, namely of 2.6 ns coupled to Histones in the nucleus in the absence of acceptor and of 2.4 ns in the presence of an acceptor (*Llères et al., 2017*). Furthermore, despite the fact that all junctions are labelled with the E-cad Alexa 555 acceptor (see *Figure 2—figure supplement 1b*), modulations in the labelling efficiency may be at the origin of variations observed in the lifetime. In addition, it is not possible to interpret the observed mean increase of lifetime, which is in the order of ~0.1 ns in all FLIM experiments, in terms of β-cat release. The reason is that the released fraction of free β-cat in the cytoplasm should escape detection and not lead to a higher lifetime value due to an increased population of GFP in a no-FRET configuration (see *Figure 2—figure supplement 2*). Indeed, the free fraction released should necessarily be at much of 100% with a strong dilution from a $2\pi r$ junctional line into the $\pi r^2 l$ cell volume, where r = 2.5 μm is the cell apex radius and l = 15 μm is the cell length. Hence, the contribution of the free fraction of $100\% * 2\pi r * /\pi r^2 l * d^2 = 5.10^{-6}$ is negligible, with $d^2 = (10\ nm)^2$ being the cross-section surface of the molecular complex.

These experiments thus demonstrate the existence of a mechanical deformation involving a specific site connecting two interacting proteins. The molecular detail of this deformation involves the stretching of the β-cat and E-cad helices which interact through their respective Y654 and D665 residues, during mesoderm invagination.

It is important to confirm that the mechanical strain at the molecular level around the Y654-D665 interacting sites during mesoderm invagination is caused by the mechanical strain developed at the cell tissue level by the invaginating cells of the mesoderm tissue (*Costa et al., 1993*). For this purpose, we first evaluated the FLIM signal in Snail (Sna) and Twist (Twi) deficient embryos that are defective in mesoderm invagination. Snail and Twist were knocked-down by injecting *sna* and *twi* siRNAs for RNAi (*Mitrossilis et al., 2017*). We then rescued the mechanical strains that are lacking in the *sna twi* RNAi mesoderm invagination defective embryos. This was achieved by injecting ultra-magnetic liposomes (UMLs) into mesoderm cells (*Brunet et al., 2013*; *Mitrossilis et al., 2017*) and applying a magnetic field gradient tangential to the mesoderm, and perpendicular to its antero-posterior axis (*Figure 3a* and Method section). This gradient resulted in a force that constricts cells laterally, with a deformation of the apexes which is similar to the uniaxial apical constriction generating mesoderm invagination (*Figure 3b,c*) (*Sweeton et al., 1991*).

We found no significant increase in the FRET lifetime between stage 6 (stage of invagination in the WT) and stage 5 in *sna twi* RNAi embryos defective in mesoderm invagination (*Figure 3d*, *Figure 3—source data 1*, without magnet injected with UMLs and siRNAi against *twi* and *sna*), in contrast to the increase observed in the WT (*Figure 2c*). This observation shows a defect of mechanical strain deformation between the linear domain II of E-cad and the C-terminus of β-cat induced in *sna twi* embryos defective in mesoderm invagination. In response to the magnetically induced constriction, junctional FRET live imaging showed an increase of the lifetime, observed by an increase of bright green and of yellow spots in junctions of the deformed tissue (e.g. yellow arrows in *Figure 3b*), of 0.098 ± 0.057 ns, comparable to the increase of 0.094 ± 0.059 ns observed in the mesoderm during its invagination. This lifetime increase shows an increase of distance between the two fluorophores on the order of ~1 nm (*Figure 3d,e* and Materials and methods). As controls, no increase in the lifetime between stage 5 and stage 6 was observed after injection of the UMLs and RNAi without magnet (*Figure 3d*), or with magnets after UML and PBS injection without the E-cad Alexa 555 acceptor (*Figure 3d*).

Thus, the defect of molecular mechanical strain deformation between the fluorescently labelled linear domain II of E-cad and the C-terminus of β-cat in *sna twi* RNAi embryos is rescued mechanically. Consequently, it is associated to the lack of mechanical strains in mesoderm defective embryo. These observations demonstrate that the mechanical strains developed by mesoderm invagination induce a mechanical stretching deformation in the order of ~1 nm in the molecular hetero-complex between the HII-E-cad and H3-β-cat helices, which are linked by the Y654-D665 site. These results confirm quantitatively the 1.25 ± 0.6 nm deformation predicted in our MD simulations.

We then tested if the increase of endogenous mechanical strain stretching the HII-E-cad and H3-β-cat helices causing the Y654-D665 binding site to open, leads to the increase of accessibility of

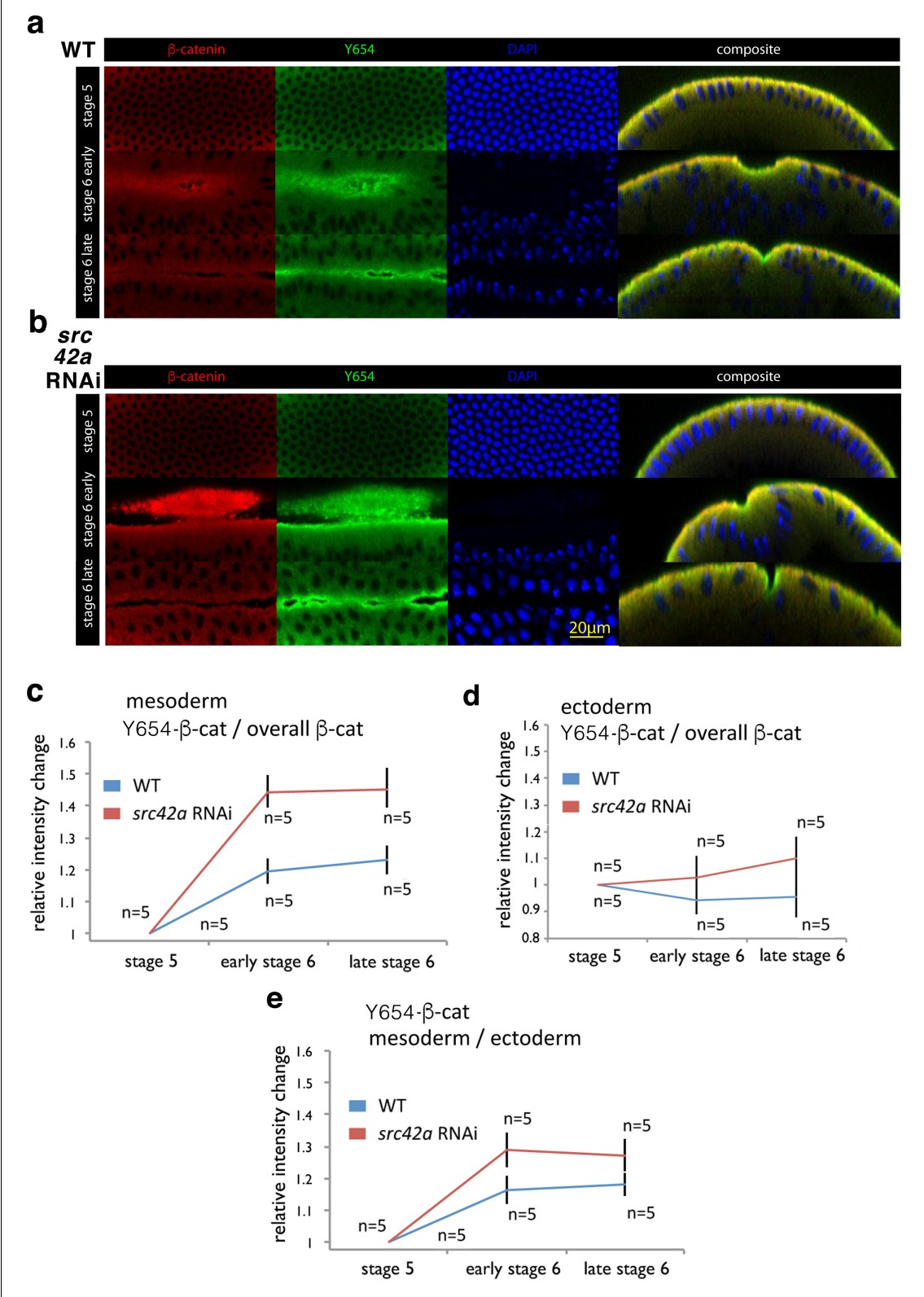

**Figure 4.** Increase of accessibility of Y654-β-cat in the mesoderm during invagination. (**a, b**) Fluorescent labelling of wild type (WT) (a) and *src42A* RNAi (b) embryos at three different stages for β-cat in red, the not phosphorylated tyrosine Y654 of β-cat in green and nuclei in blue (DAPI). Composite are cross-cuts of the 3D reconstructed images of the latter three colour images of the embryo. (**c**) Relative intensity change of antibody fluorescent signal against the non-phosphorylated non-phosphorylated Y654-β-cat site of Src42A phosphorylation in the mesoderm junctions, normalized to the junctional

*Figure 4 continued on next page*

*Figure 4 continued*

total β-cat signal, in wild type (WT) and *src42A* RNAi (Src42A knock down) background condition. Values are relative to stage 5. N = 5 in the WT, and N = 5 in the *src42A* RNAi. Mean and SEM are shown, t-test p=value 0.00181 and 0.00021 at early stage 6, 0.000508 and 0.00330 at late stage 6 for WT and *src42A* RNAi, respectively. WT compared to *src42A* RNAi early stage 6 p=0.00355, late stage 6 p=0.00879, difference between the two curves p=0.0003. (**d**) Relative intensity change of antibody fluorescent signal against the non-phosphorylated Y654-β-cat site of Src42A phosphorylation in the ectoderm junctions, normalized to the junctional total β-cat signal, in wild type (WT) and *src42A* RNAi (Src42A knock down) background condition. Values are relative to stage 5. N = 5 in the WT, and N = 5 in the *src42A* RNAi. Mean and SEM are shown, t-test p=value of 0.439 and 0.738 at early stage 6, of 0.578 and 0.347 at late stage 6 for WT and *src42A* RNAi, respectively. WT compared to *src42A* RNAi early stage 6 p=0.247, late stage 6 p=0.162, difference between the two curves p=0.169. (**e**) Relative intensity change of the non-phosphorylated Y654-β-cat site of the mesoderm signal to the non-phosphorylated Y654-β-cat ectoderm signal within the same embryos. Mean and SEM are shown, t-test p=value 0.0103 and 0.000278 at early stage 6, 0.0125 and 0.00256 at late stage six for WT and *src42A* RNAi, respectively. WT compared to *src42A* RNAi early stage 6 p=0.049, late stage 6 p=0.063, difference between the two curves p=0.021.

DOI: https://doi.org/10.7554/eLife.33381.017

The following source data is available for figure 4:

**Source data 1.** Increase of accessibility of Y654-β-cat in the mesoderm during invagination.
DOI: https://doi.org/10.7554/eLife.33381.018

Y654, as predicted by the opening of the site, with a probability of 15% by MD. To do so, we made use of an antibody directed against the non-phosphorylated Y654 site in the β-cat protein (Y654-Ab). To prevent disturbing competition between the Y654-β-cat antibody and E-cad on the Y654 site by injection of the E-cad anti-body *in vivo* before gastrulation, immuno-fluorescent staining was performed on fixed embryos using the ultra-rapid MeOH heat procedure that maintains junctional labelling only (*Müller and Wieschaus, 1996*), at stage 5, early stage 6 and late-stage 6 embryos. The apical punctual nature of the labelling can be observed at the onset of gastrulation, as curvature is still low enough to prevent compacting apical junctions together.

We found an increase in the signal for the un-phosphorylated site of β-cat at AJ relative to the junctional β-cat signal during gastrulation in the invaginating tissue. The increase is specific of gastrulation early stage 6 and late-stage 6 compared to pre-gastrulation stage 5, of 19.6 ± 6.2% and 23.0 ± 5.7% respectively (*Figure 4a,c*, *Figure 4—source data 1*). As a control, no increase in the signal for the un-phosphorylated site of β-cat at AJ relative to the junctional β-cat signal observed in the ectoderm, which is less strained (*Figure 4d*, *Figure 4—source data 1*). Consistently, an increase of 16.5 ± 7.5% and 18.0 ± 5.5% respectively, is observed in the signal for the un-phosphorylated site of β-cat at AJ in the mesoderm normalized to the un-phosphorylated site of site of β-cat in the ectoderm (*Figure 4e*, *Figure 4—source data 1*). This result thus shows that under the molecular strain induced by the morphogenetic movement of invagination, the Y654 site increases its accessibility by nearly 20 ± 5% to specific molecular interactions, before being phosphorylated. This is in line with our quantitative predictions by the model, and based on the simulation of the Y654-D665 binding site opening of 15% under strain.

The mechanically induced opening of the Y654-D665 binding site should thus favour the accessibility of the Y654 tyrosine to Src42A and thereby its phosphorylation by Src42A which leads to the release of the β-cat from AJ (*Brunet et al., 2013*). However, such Src42A interaction with the Y654 site might be difficult to detect, as it should be highly transient, and thus would yield most likely poor FRET signals by probing any increase of the accessibility effect. On the other hand, Src42A would compete with Y654-Ab in its interaction with Y654. The increase of accessibility of the Y654 site to Y654-Ab under strain should thus be enhanced in the absence Src42A. An alternative prediction of our model, leading to the same conclusion, concerns the knock down of the responsible kinase Src42A that reduces Y654-β-cat phosphorylation (*Takahashi et al., 2005*). The knock down should thus lead firstly to stable and consequently enhanced molecular stretching effects during gastrulation, and thus to an increased number of stretched Y654 sites opened to non-phospho Y654 labelling. Both effects should lead to an enhancement of the accessibility of the Y654 sites.

To test such predictions, we thus measured the Y654 accessibility to the non-phospho antibody in *src42A* RNAi embryos (*Desprat et al., 2008*). In the *src42A* RNAi knockdown background, we found a specific increase in the signal for the un-phosphorylated site of β-cat at AJ under strain, as observed by the increase of Y654-βcat green labelling that adds to, and dominates the β-cat red labelling, leading to a bright green/yellow composite colour (*Figure 4b*).

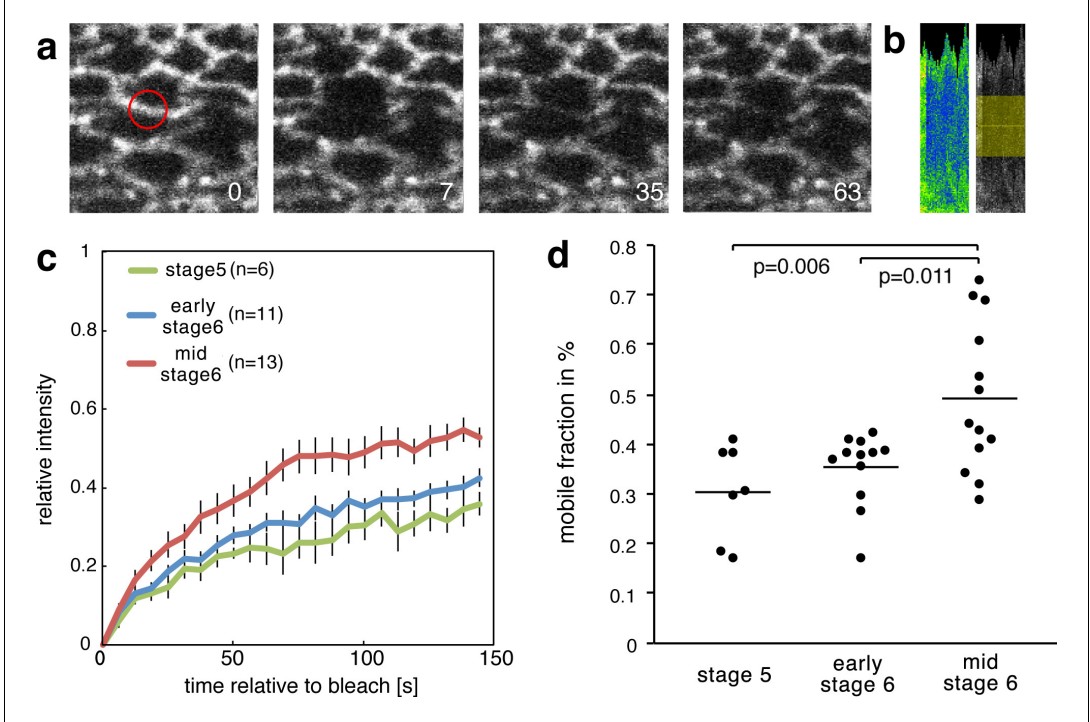

**Figure 5.** Increase of the mobile fraction of β-cat-GFP in the mesoderm during invagination. (**a**) Image sequence of mesodermal cells expressing β-cat -GFP during a FRAP experiment. The red circle shows the region of photobleaching. Time in seconds is given relative to bleaching. (**b**) Corresponding kymograph of d in rainbow LUT and black and white with indicated ROI for value calculation. (**c**) Relative intensity after photobleaching of β-cat-GFP in the mesoderm at three different stages during early *Drosophila* embryonic development, just before and during invagination. Time in seconds is given relative to bleaching. (**d**) Mobile fraction in %. Mann-Whitney test p-values are p=0.011 and p=0.006 for early stage 6 (N = 12) and mid stage 6 (N = 13) compared to stage 5 (N = 7).

DOI: https://doi.org/10.7554/eLife.33381.019

The following source data and figure supplement are available for figure 5:

**Source data 1.** Increase of the mobile fraction of β-cat-GFP in the mesoderm during invagination.
DOI: https://doi.org/10.7554/eLife.33381.021
**Figure supplement 1.** Synthetic view of the findings (box) integrated in the current knowledge.
DOI: https://doi.org/10.7554/eLife.33381.020

This signal was measured relative to the junctional β-cat signal during gastrulation (44.3 ± 7.3% and 45.1 ± 7.9%, respectively), and of about 25 ± 5% compared to wild type, both at early stage 6 and late-stage 6 (*Figure 4b,c*). As a control, no significant increase in the signal for the un-phosphorylated site of β-cat at AJ relative to the junctional β-cat signal is observed in the ectoderm, that is less strained, in the *src42A* RNAi embryos compared to the WT (*Figure 4d*). Consistently, an increase of 12.4 ± 5.8% and 9.3 ± 4.3% respectively, is observed in the signal for the un-phosphorylated site of site of β-cat at AJ in the mesoderm normalized to the un-phosphorylated site of β-cat ectoderm, in the *src42A* RNAi embryos compared to the WT (*Figure 4e*). These results showed the increase of accessibility of Y654 to Y654-Ab of nearly 20% in the absence of Src42A under strain.

Therefore, we conclude that during mesoderm invagination, the accessibility of the site of Y654 β-cat to Src42A increases due to its mechanical opening. At the same time, the Y654 β-cat residue becomes specifically more available for Src42A phosphorylation at AJ, as the strain builds up at the Y654 β-cat-D665-E-cad molecular complex interaction site. The mechanical induction of the increase in accessibility of the Y654 site under strain for Src42A phosphorylation should thus lead to the release of an amount of junctional β-cat into the cytosol proportional to the subset of β-cat/E-cad complex that opens at Y654-D665 under strain (*Brunet et al., 2013*; *Roura et al., 1999*; *van Veelen et al., 2011*).

To test whether the molecular strain that is observed in mesoderm invagination induces a decrease in β-cat affinity to E-cad, we characterised the dynamics of β-cat in *Drosophila* embryo mesodermal AJs under strain during gastrulation. This was achieved by fluorescent recovery after photobleaching (FRAP) experiments at three different developmental stages: just before invagination (end of cellularization, late embryonic stage 5), at the onset of invagination (early 6) and during invagination (mid-stage 6) (*Figure 5a,b*).

We found upon photobleaching that the fluorescent recovery amplitude, the so-called mobile fraction, of β-cat upon photobleaching is increased by 16 ± 6% when mesoderm invagination has built up at mid-stage 6. This observation indicated that about 16% more molecules at the AJ are able to be recycled and exchanged at stage 6 compared to stage 5 (*Figure 5c,d*, *Figure 5—source data 1*). This result is consistent with studies examining β-cat turnover at cellular junctions under strain during germ band extension (*Tamada et al., 2012*). This effect is quantitatively predicted by the 15% probability of opening the β-cat/E-cad complex observed in our MD simulations under strain. As the opening of the Y654-D665 interacting site will lead to an increased Src42A phosphorylation, it will therefore lead to a decreased affinity between β-cat and E-cad.

This relationship shows that the morphogenetic movements of invagination locally cause a mechanical stretch at the Y654-D665 interaction site in the β-cat/E-cad molecular complex. It indeed statistically weakens the interaction of β-cat with E-cad in AJ of the mesoderm cells and thereby favours cytosolic β-cat enrichment.

## Discussion

In the gastrulating *Drosophila* embryo, the Src42A kinase required for the mechanical induction of Y654-β−cat phosphorylation is already activated (phosphorylated on Y400) before tissue deformation. Therefore, Src42A is surprisingly not mechanically 'over-activated' by tissue deformation during the morphogenetic movements of gastrulation (*Desprat et al., 2008*), and is thus not upstream of the mechanotransduction pathway leading to Y654-β-cat phosphorylation. In contrast, the already activated form of Src42A is permissively required in the mechanotransductive process leading to the activation of β-cat release from the junctions into the cytoplasm and then to the nucleus. This observation opened the question of the underlying molecular mechanism of the mechanotransductive process leading to the β-cat mechanical activation, which is conserved and generically involved in many diverse physiological and pathological biological conditions (*Fernandez-Sanchez et al., 2015a*; *Farge, 2003*; *Desprat et al., 2008*; *Brunet et al., 2013*; *Benham-Pyle et al., 2015*; *Fernández-Sánchez et al., 2015b*; *Samuel et al., 2011*; *Whitehead et al., 2008*; *Benham-Pyle et al., 2016*; *Sen et al., 2008*; *Shyer et al., 2017*) and suggests the Y654 β-cat interaction site with E-cad as the underlying molecular mechanosensor.

Here we showed that the highly evolutionary conserved Y654 β-cat-D665-E-cad major interaction site of the AJ complex acts as primary mechanosensor. The site activates the β-cat pathway through its mechanical opening *in vivo* that enables phosphorylation by Src42A kinase. Phosphorylation in turn leads to the decrease of the Y654 β-cat-D665-E-cad affinity (*Roura et al., 1999*; *van Veelen et al., 2011*), and to the release of β-cat into the cytosol. This release finally leads to target developmental patterning gene transcription in the nucleus (*Figure 5—figure supplement 1*) (*Brunet et al., 2013*).

We thereby demonstrate *in vivo* the existence of a primary molecular mechano-chemical translator. It is functionally involved in the trans-scale activation of a molecular physiological differentiation pathway, in response to an endogenously produced macroscopic mechanical strain. Here, the mechanosensor is the Y654-β-cat/D665-E-cad interaction site of the hetero β-cat/E-cad complex at AJ. It opens under strain and favours the phosphorylation of β-cat by Src42A. This process initiates early mesoderm differentiation and specification in response to the strain associated to mesoderm invagination at the onset of *Drosophila's* embryonic gastrulation. The mechanical activation of the β-cat pathway shows a fine regulation under physiological strains. The soft stretching of 1 nm between the two α−helices surrounding the Y654-β-cat/D665 E-cad major interaction site leads to a probability of 15% opening and release of signalling β-cat from the junctions after phosphorylation by Src42A. Such finely tuned process could trigger the mechanotransductive signalling translocation from AJ to the nucleus of low levels of β-cat, known to be sufficient to trigger transcription into the nucleus (*Peifer and Wieschaus, 1990*), and ensure the maintenance of tissue coherence by the

maintenance of 85% of the β-cat in the junctions. This scenario would explain the high sensitivity of β-cat target gene expression to low levels of β-cat expression (*Peifer and Wieschaus, 1990*). A possible interpretation for the observed 15% efficiency of site opening is that, once under tension, the deformation of the β-cat/E-cad complex can *a priori* statistically occur at different locations all along the complex structure. In 15% of cases, the deformation would occur on the effective Y654-β-cat/D665-E-cad site and open it. Future molecular dynamics investigations will be required to quantitatively test this hypothesis.

Finally, both the Y654-β-cat and D665 E-cad major sites between the two molecules are strongly conserved across all metazoa (*Figure 1—figure supplement 3*). This mechanotransductive molecular sensor may thus be generic in the overall animal kingdom, and be involved in physiological mechanical induction processes, including mesoderm differentiation across all bilaterians (*Brunet et al., 2013*), and pathological processes, including mechanical induction of tumour differentiation in response to tumour growth pressure in most epithelia of most animals (*Fernández-Sánchez et al., 2015b*).

# Materials and methods

**Key resources table**

| Reagent type (species) or resource | Designation | Source or reference | Identifiers | Additional information |
|---|---|---|---|---|
| Antibody | Anti-Y654-b-cat | Sigma-Aldrich | RRID:AB_10623284 | |
| Antibody | Anti-Arm | DSHB | RRID:AB_528089 | |
| Antibody | JCAb20 | abcam antibody service | | against H- CALQGNMRDPNDIPDI - NH2 (16AA) corresponding to AA 1143–1158 of the DE-cad |
| Antibody | Anti-DE-cad DCAD2 | DSHB | RRID:AB_528120 | |
| Strain, strain background (*Drosohila melanogaster*) | Arm-GFP | Bloomington Drosophila Stock Center | RRID:BDSC_8556 | |
| Strain, strain background (*Drosohila melanogaster*) | mata4-GALVP16/V37 tubulin | Bloomington Drosophila Stock Center | RRID:BDSC_7063 | |
| Strain, strain background (*Drosohila melanogaster*) | Oregon-R | Bloomington Drosophila Stock Center | RRID:BDSC_5 | |
| Strain, strain background (*Drosohila melanogaster*) | *src42A* RNAi | NIG Stock Centre | 7873 R-2 | The NIG Stock Centre is not yet in the RRID system |
| Strain, strain background (*Drosohila melanogaster*) | UAS-*src42A*-RNAi (II) | VDRC Stock Centre | RRID:FlyBase_FBst0452809; 17643 | |
| Sequence-based reagent | *sna*-siRNAs | Produced by Eurogentec | | Sequences given in methods |
| Sequence-based reagent | *twi*-siRNAs | Produced by Eurogentec | | Sequences given in methods |
| Software, algorithm | FLIMfit 4.12.1 | Imperial College London | | |
| Software, algorithm | SymphoTime | Leica/Picoquant system | | |
| Software, algorithm | Packing Analyzer | Benoit Aigouy | | doi:10.1016/j.cell.2010.07.042 (2010) |
| Commercial assay or kit | Alexa Fluor 555 Antibody Labeling Kit | Molecular probes lifetechnologies | Cat. No. A20187 | |
| Commercial assay or kit | Antibody concentration kit | abcam | Cat. No.102778 | |
| Chemical compound, drug | Ultra-magnetic liposomes (UML) | | | doi:10.1021/la3024716 (2012). |

## Simulations

We used the NAMD 2.10 software on GPU/CPU nodes to run the molecular dynamics (MD) simulations in explicit solvent (*Phillips et al., 2005*). The CHARMM36 forcefield was employed for the protein, together with the TIP3P-CHARMM water model (*Huang and MacKerell, 2013*). The system was neutralized with sodium and chloride ions. Starting from the crystal structure of the complex (pdb id: 1I7X) that contained an incomplete dimer of the complex, we selected the chain A (for β-cat) and chain B (for E-cadherin), completing a missing loop in chain A between residues 553 and 560 with the Modloop online software (https://modbase.compbio.ucsf.edu/modloop/). Chain B was truncated at residue 648 on the N-terminal side, as the corresponding portion of the chain would be extended *in vivo* and not interacting with β-cat (as observed in the crystal structure). The system was solvated in a water box of 7.2 * 7.2 * 21 nm, so that the protein long axis (defined by the two terminal alpha carbons of β-cat) was aligned with the z direction. The system was then minimized using a steepest-descent algorithm, and was progressively heated to 300 K during 120 ps. The bond between Y654 and D665 was constrained to its value in the crystal structure to ensure that the heating toward room temperature did not dissociate the complex. The system was then equilibrated for 2 ns at ambient pressure (1 bar) and temperature (300 K) using Langevin thermostats and barostats, and keeping the constraint between the two residues. The constraint was then progressively relaxed during 2 ns, and the system was subsequently equilibrated for 10 ns under the same conditions. 10 control trajectories in the absence of force were propagated for 30 ns. To ensure that the protein complex stayed aligned to the z axis in the elongated simulation box, and could not interact with its images by rotating around the z direction, a light harmonic constraint (force constant = 5 kcal/mol) was applied to the projection of the end-to-end axis of β-cat in the (xy) plane. This constraint did not affect the interaction between β-cat and E-cad, as we verified that the same simulations (but for a shorter simulation time of 10 ns so that the protein could not significantly tumble) in the absence of such constraint lead to quantitatively similar results. For the simulations under force, we fixed the alpha carbon of Y142 of β-cat, and we applied a constant force of 150 pN in the z direction to the alpha carbon of A648 (i.e., the N-terminal) of E-cad. 10 simulations were propagated for 30 ns. Analysis of interatomic distances were performed using in-house Fortran 90 codes, and visualization was done using the VMD software.

## Fly strains

Stocks were maintained at room temperature (typically 22°C). Oregon-R (used for WT), Mat-Gal4 (III) (mata4-GALVP16/V37 tubulin, RRID:BDSC_7063) and β-cat-GFP (arm-GFP, RRID:BDSC_8556) were provided by Bloomington. Homozygous UAS-*src42A* RNAi was produced by crossing 7873 R-2 *src42A* RNAi with the 17643 UAS-*src42A*-RNAi (II) (RRID:FlyBase_FBst0452809) provided by the NIG Stock Centre and VDRC Stock Centre, respectively. UAS*Gal4 crosses were stored 2 hr at 28°C prior to experiments.

## Immunostainings

Embryos were dechorionated and fixed according to Muller, H. A. and Wieschaus J. Cell Biol. 134, 149–163 (1996) (*Takahashi et al., 2005*). Proteins were detected with the following primary antibodies: rabbit anti-Y654-β-cat, Sigma-Aldrich (dilution 1:200) RRID:AB_10623284; mouse anti-Arm (*Drosophila* β-cat) antibody RRID:AB_528089, DSHB (dilution 1:200). Secondary antibodies labelled with Alexa 488 or Alexa 633 were used from Invitrogen (dilution 1:200). Embryos were mounted vertically in Vectashield with DAPI for confocal observation. Images were obtained with a Zeiss 510 inverted confocal microscope. Images were processed and analysed with Fiji software. The intensity of the non-phospho-Y654 antibody staining in the mesoderm was either normalised to the β-cat intensity within the same ROI or to the intensity of the non-phospho-Y654 antibody staining in the ectoderm, all after background substraction (measured outside the embryo).

## Western blots

Western blots were conducted as described in D Mitrossilis et al. Nat Commun 8, 13883. 2017 Jan 23 with the following specifications: the JCAb20 was used in a 1:500 dilution and the anti-DE-cad DCAD2from *Drosophila* Hybridoma bank was used in a 1:500 dilution, RRID:AB_528120.

## Immunoprecipitation procedure

70–80 β-cat-GFP embryos at stage five were lysed in 50 mM Tris pH 8.0, 150 mM NaCl, 1% Triton X-100, 1 mM MgCl2 buffer plus a protein inhibitor cocktail (Sigma). After centrifugation at 14000 rpm for 20 min, protein lysate was pre-cleared with 50 μl of protein A-agarose beads (Cell Signaling Technology), incubated with 10 μg of the JCAb20 antibody over night at 4°C, and subsequently incubated with 50 μl of protein A-agarose beads for 3 hr at 4°C. After washing, bound proteins were eluted with gel sample buffer and analyzed by Western-blot using the anti-DE-cad DCAD2 antibody (dilution 1:500).

## RNAi

To produce *twi* and *sna* defective non gastrulating Arm-GFP embryos, we injected siRNAs for *twi* and *sna* RNAi into Arm-GFP embryos. *twi sna* RNAi embryos have been described to phenocopy *twi sna* double zygotic mutants, and do not gastrulate (*Mitrossilis et al., 2017*; *Martin et al., 2009*). 3 siRNA sequences were designed and produced by Eurogentec, and already used in reference (*Mitrossilis et al., 2017*):

*sna*-siRNAs (1481–1503 s 5': CAGCUAUAGUAUUCUAAGU dTdT; AS 3': ACUUAGAAUACUA UAGCUG dTdT, 340–362 s 5': GGAACCGAAACGUGACUAU dTdT; AS 3': AUAGUCACGUUUCGG UUCC dTdT, 972–994 s 5': CCGAGUGUAAUCAGGAGAA dTdT; AS 3': UUCUCCUGAUUACAC UCGG dTdT)

*twi*-siRNAs (928–950 s 5': GCACCAGCAACAGAUCUAU dTdT; AS 3': AUAGAUCUGUUGCUGG UGC dTdT, 2061–2083 s 5': GUCACGCUUUCCAUAUAUA dTdT; AS 3': UAUAUAUGGAAAGCG UGAC dTdT; 1–1993 s 5': CGGAUCAGGACACUAUAGU dTdT; AS 3': ACUAUAGUGUCCUGA UCCG dTdT)

## Injection of UML and induction of RNAi

Conditions of injection with the UMLs and siRNA are described in reference (*Mitrossilis et al., 2017*).

An injector (Eppendorf Femtojet) was fixed on the microscope to inject magnetic particles or/and siRNA. A x,y,z piezo-electric system (Princeton Instruments) was mounted on the microscope to control the position of the embryo mounted on a coverslip relative to the injecting needles, the micro-magnets, or the indent needle, with the 100 nm resolution.

*UML injections.* Using a Femtojet Eppendorf injector ~0.05 μl of 0.1 μm/g UML were injected into the ventral cells and the basal domain of the ventral cells of the embryo from 20 to 40 min before the end of ventral cellularisation. Injections were performed at constant pressure, with the visualisation of UML labelled with Rhodamine observed in red in spinning disc. Prior to initiation of the *sna*-dependent pulsating phase, all blastoderm cells are under cellularisation and are basally open to the yolk. Ultra-magnetic liposomes (UML), composed of a high concentration of 9 nm Fe-oxide superparamagnetic nanoparticles (20% v:v) encapsulated in 200 nm fluorescently labelled phospholipid membranes (*Béalle et al., 2012*), were thus injected into the yolk below and all along mesoderm cells, 20 to 40 min before the end of cellularisation. Note that WT embryos injected with UML gastrulated normally.

*UML and siRNA Injections.* ~ 0.05 μl of 0.1 μm/g UML and ~0.05 μl of 0.1 μg/μl siRNA were injected from 80 to 60 min before the end of ventral cell cellularisation. The UML and the siRNA were injected together inside the embryo and the basal domain of the ventral cells.

## Magnetic deformation device

A 2D dense network of 5 microns hard permanent NdFeB micro-magnets pillars separated by five microns embedded into pdms (symbolized by a magnet in *Figure 3a*), was approached tangentially to the UML injected mesoderm of the embryo, at a distance of 10 microns. At this distance, the pillars produce a mean magnetic field of 400mT with a magnetic field gradients on the order of $10^4$ T. $m^{-1}$ (*Roy et al., 2016*), which was sufficient to mimic the earliest stages of apex constriction that initiate mesoderm invagination. (*Figure 3b*).

## FRAP experiments

FRAP experiments were performed on an inverted confocal Zeiss 510 microscope. Arm-GFP expressing embryos were imaged using the 488 nm laser line of the microscope. Bleaching was performed using the 488 nm and 514 nm laser lines at 100% for three seconds. Recovery of bleached areas was monitored with a frame rate of 6 s for 3 min after bleaching. Cell boundaries were identified and kymographs of bleached boundaries were produced using Packing Analyzer v2.0 (*Aigouy et al., 2010*). To correct for photobleaching due to image acquisition, we measured the average intensity of all unbleached cell boundaries I in each frame n of the movie and normalized average intensities to the intensity of the first frame ($I_0$). This results in a correction factor C with $C_n = I_0/I_n C_n = I_0$. Bleached cell boundaries were tracked and kymographs were created using Packing Analyzer.

## FRET experiments

FRET experiments were performed on an inverted 2-photon Leica TSC SP8 NLO SMD (Picoquant module) microscope system or on a spinning disc CSU10 head (Yokogawa) mounted on a Nikon TE2000 microscope, both equipped with a Lifa module (Lambert Instruments) by measuring the lifetime of the donor, so called FLIM-FRET. The FRET efficiency was thereby measured as a shortening of the donor lifetime. Lifetime measurements were analysed using either the SymphoTime software (Leica/Picoquant system) or the FLIMfit 4.12.1 (software tool developed at Imperial College London), with a monoexponential fit. For this, the lifetime of the donor alone was first determined for calibration of the system. Afterwards the lifetime of the donor in the presence of the acceptor in the mesoderm and ectoderm region was analysed at the level of the adherens junctions in the lifetime images.

Fluorescently labelled antibodies were injected into the *Drosophila* embryo before cellularization using an Eppendorf Transjector as described in Misquitta L et al (*Misquitta et al., 2008*). with the following alterations. Ready-made injection needles from Eppendorf were used and a self-build micromanipulator mounted on an Olympus IX70 microscope. Roughly 0.05 µl of the antibody solution were injected for a 1:200 dilution of the antibody in the embryo (a *Drosophila* embryo is W = 0.18 ± 0.001 mm wide and L = 0.51 ± 0003 mm high according to Markow TA et al (*Markow et al., 2009*), which gives an approximate volume of 9.02 ± 0.14 mm$^3$ or 9 µl using the formula $(1/6)\pi W2L$)

## Primary antibody design for FRET

The antibody was purified from rabbit that was immunized with the following peptide: H-CALQG NMRDPNDIPDI-NH2 (16AA) corresponding to the AA 1143–1158 of the DE-cad protein in Drosophila. The sequence was chosen based on the position of it's corresponding mouse sequence in the crystal structure of β-cat in complex with the cytoplasmic part of the E-cad protein (PDB 1i7x). The DE-cad sequence corresponds to a loop in Region II adjacent to the alpha-helical stretch that contains the D1170 (D665 mice), which interacts with the Y667 (Y654 mice) phosphorylation site of β-cat in the complex. Peptide synthesis and antibody production was carried out by eurogentech. Received antibody solutions were stored in in glycerol (1:2) at −20°C. The antibody was concentrated using an antibody concentration kit from abcam, labelled with Alexa 555 using the Alexa Fluor 555 Antibody Labeling Kit from lifetechnologies and stored in PBS at 4°C.

## Calculation of the strain from FRET efficiency variations

The FRET efficiency was calculated using the E = 1 tda/td with tda being the lifetime of the donor in the presence of the acceptor, while td denotes the lifetime of the donor alone. The distances r are then calculated as r = R0((1/E)−1)ˆ1/6 with R0 being the Förster Radius, in the approximation that there are no cross-FRET between adjacent molecules (*Wallrabe and Periasamy, 2005*). The Förster Radius for this particular FRET pair (GFP and Alexa 555) is R0 = 6.3 nm (Thermofisher datasheet).

## Statistics

A sample is one embryo or simulation, and one measurement is associated to one embryo or one simulation. N is the number of embryos or simulations by Figure, so N is the number of time the experiment was replicated both biologically and technically. Experiments and simulations are

realized one by one on single embryos and single molecular complexes: the N sample size was thus designed *a posteriori* once the p-value showed statistically significant results enough, relative to the difficulty and time consuming of individual experiments and simulations. Statistics are based on the ANOVA test (when allowed by the Shapiro-Wilk's normality test and in case standard deviations are similar between distinct conditions (*Ananda and Weerahandi, 1997*), which is the case for *Figure 2c* only), Mann-Whitney test, and the t-student test when allowed by the Shapiro-Wilk's normality test, two-sided. P values for comparing two curves were calculated following reference (*Fisher, 1932*) with Rstudio (see 'pvalue Fisher t-test curves comparison' *Source code 1*).

Quantitative analysis was systematically done by automatic microscope image analysis or simulation dedicated algorithms, within the exact same conditions for all control/perturbed sample data in series, independently of the unmasked group allocation done by the investigator.

## Acknowlegements

This research was funded by the ANR (grant # 11 BSV5014-01), the AVIESAN Cell Biology, Development and Evolution (CNRS, INSERM, INSTITUT CURIE co-funding of the dedicated Magneto-Spinning disk), and the Labex DYNAMO (ANR-11-LABX-0011–01). EF lab is funded by the FRM (grant # DEQ20150331702). JCR was funded by the Institut Curie post-doc starting grant, the Marie Sklodowska Curie Fellowship (PIEF-GA-2012–332422), the Labex CelTisPhyBio (post-doc grant # 11-LBX-0038). The authors are very grateful to Christine Ménager for providing the UMLs required for magnetic manipulation. The PICT-Cell and Tissue Imaging Facility is a member of the French National Research Infrastructure France-BioImaging (ANR-10-INBS-04).

## Additional information

### Funding

| Funder | Grant reference number | Author |
| --- | --- | --- |
| Agence Nationale de la Recherche | 11 BSV5014-01 | Jens-Christian Röper<br>Démosthène Mitrossilis<br>Emmanuel Farge |
| Agence Nationale de la Recherche | 11-LBX-0038 | Jens-Christian Röper<br>François Waharte<br>Jean Salamero<br>Emmanuel Farge |
| European Commission | PIEF-GA-2012-332422 | Jens-Christian Röper |
| Agence Nationale de la Recherche | ANR-11-LABX-0011-01 | Guillaume Stirnemann<br>Marc Baaden |
| Agence Nationale de la Recherche | ANR-10-INBS-04 | Jean Salamero |
| Fondation pour la Recherche Médicale | DEQ20150331702 | Emmanuel Farge |

The funders had no role in study design, data collection and interpretation, or the decision to submit the work for publication.

### Author contributions

Jens-Christian Röper, Conceptualization, Data curation, Formal analysis, Investigation, Visualization, Methodology, Writing—original draft, Designed experiments (with EF), Realized experiments on FLIM controls and in the presence of magnetic forces (with DM) and immunoprecipitation (with MEFS); Démosthène Mitrossilis, Data curation, Investigation, Methodology, Set up magnetically induced mechanical deformation experiments; Guillaume Stirnemann, Data curation, Software, Formal analysis, Investigation, Visualization, Methodology, Designed and generated numerical molecular simulations; François Waharte, Resources, Data curation, Formal analysis, Methodology, Designed FRET experiments (with JCR and JS); Isabel Brito, Software, Wrote the R script to compute the Fisher (1932) theory for p-values of combined p-values allowing the calculation between two

curves; Maria-Elena Fernandez-Sanchez, Investigation, Performed the immunoprecipitation experiment and produced FLIM controls (with JCR); Marc Baaden, Resources, Validation, Supervised coordination between numerical and wetlab experiments (with EF); Jean Salamero, Conceptualization, Resources, Validation, Designed FRET experiments (with JCR and FW); Emmanuel Farge, Conceptualization, Supervision, Funding acquisition, Validation, Methodology, Writing—original draft, Project administration, Writing—review and editing, Supervised coordination between numerical and wetlab experiments (with MB), Designed experiments (with JCR), Coordinated the overall work

## Author ORCIDs
Démosthène Mitrossilis (iD) https://orcid.org/0000-0003-3382-296X
Marc Baaden (iD) https://orcid.org/0000-0001-6472-0486
Emmanuel Farge (iD) http://orcid.org/0000-0002-5063-7179

## Decision letter and Author response
Decision letter https://doi.org/10.7554/eLife.33381.027
Author response https://doi.org/10.7554/eLife.33381.028

# Additional files

## Supplementary files
• Source code 1. 'pvalue Fisher t-test curves comparison'. Entering the values in the 'experiment.xls' file ('1' is time 1 and '2' is time 2 of the experimental curve, C is control and P is perturbed conditions), saving it and putting it in the same file than the script, opening the script under Rstudio on MacOS10, selecting the program from line 1 to line 83, running it, selecting the test 't' line 85, running it, leads to the p-value.
DOI: https://doi.org/10.7554/eLife.33381.022

• Transparent reporting form
DOI: https://doi.org/10.7554/eLife.33381.023

## Data availability
All data generated or analysed during this study are included in the manuscript and supporting files. Source data files have been provided for: Figure 1c,d; Figure 1-supplementary; Figure 2b; Figure 2c; Figure 3d; Figure 4c,d,e; Figure 5d; Figure 2-supplementary figure 2b.

The following previously published dataset was used:

| Author(s) | Year | Dataset title | Dataset URL | Database, license, and accessibility information |
|---|---|---|---|---|
| Huber AH, Weis W | 2001 | Beta-catenin/E-cadherin complex | https://www.rcsb.org/structure/1i7x | Publicly available at the RCSB Protein Data Bank (accession no. 1I7X) |

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
