## [Decision Letter]

Thank you for submitting your article "The major β-cat/E-cadherin junctional binding site: a primary molecular mechano-transductor of differentiation in vivo" for consideration by *eLife*. Your article has been reviewed by three peer reviewers, including Reinhard Fässler as the Reviewing Editor and Reviewer #1, and the evaluation has been overseen by Vivek Malhotra as the Senior Editor.

The reviewers have discussed the reviews with one another and the Reviewing Editor has drafted this decision to help you prepare a revised submission.

Summary:

The authors describe a mechanotransduction process that induces a conformational change of a fraction of AJ-localized β-catenin leading to Src-mediated phosphorylation of β-catenin at Y654, release of pY654-β-catenin from E-cadherin, translocation into the nucleus and gene transcription. The findings are supported by molecular dynamics simulation studies and a series of in vivo experiments. These findings are significant but the manuscript lacks several important controls. Furthermore, the text of the paper is difficult to read. Sentences are fusions of several statements, figures are described in an unusual order (e.g. Figure 3E before Figure 3A), manipulations are not sufficiently explained (e.g. *sna* and *twi* RNA, not mentioned what the abbreviations mean, not described in the Materials and methods section), the Introduction is extremely brief consisting of 3 sentences followed by only 29 references. Also, the manuscript requires major editing.

Essential revisions:

1) The lifetime differences of the antibody injected embryos are surprisingly small. As it is very difficult to control the amount of antibody (and thus FRET acceptor) being present at different locations in the syncytial blastoderm one would expect much larger variations. To ensure that the reported values are reproducible the number of injected embryos should be increased (from N=8 embryos). Furthermore, an injection control is necessary, as uninjected embryos are not sufficient to control the experiment. Does water/PBS injection have an effect? Does this vary between the stages?

How was the lifetime fit done? Is it a mono-exponential fit? Is it not surprising that the GFP donor lifetime is <2.6 ns in the absence of an acceptor? This is impressively short!

2) Why are double RNAi against *twi* and *sna* injected instead of using zygotic double mutants? The small differences the authors observe upon RNAi injections could be due to differential knock-down efficiencies, particularly if the number of treated embryos is so low (N=8).

3) The rescue with the magnet should also be controlled with water injected embryos.

4) Quantification of the Y654 antibody stainings also requires a control experiment. The authors observe an increase in Y654 non-phospho antibody staining during gastrulation or upon Src knock-down. The authors seem to have normalized the Y654 non-phospho staining to general β-catenin antibody staining. An additional control, just outside of the mesoderm from the same embryo should be included, as the forces on the lateral cells should be less, the difference should also be less. Since the antibody signals can vary between embryos, such an internal standard makes it easier to compare different embryos.

5) Is it possible to design a control experiment with mutant, stretch-resistant DE-cadherin and/or β-catenin and show that if the sites are made resistant to stretching, accessibility of the Y654 site is unaffected by the application of strain?

6) The JCAb20 antibody recognises numerous bands in addition to the DE-cadherin (peptide). Based on the unspecific bindings it is not clear how valid the findings with this antibody are?

7) Do the authors have an explanation why 85% of β-catenin does not alter the association with DE-cadherin during mechanotransduction? How do these 85% of β-catenin escape the dissociation and phosphorylation by Src?

---

## [Author Response]

Summary:The authors describe a mechanotransduction process that induces a conformational change of a fraction of AJ-localized β-catenin leading to Src-mediated phosphorylation of β-catenin at Y654, release of pY654-β-catenin from E-cadherin, translocation into the nucleus and gene transcription. The findings are supported by Molecular Dynamics Simulation studies and a series of in vivo experiments. These findings are significant but the manuscript lacks several important controls.

We thank the reviewers for all of their stimulating comments and questions on the work, and for the additional important controls suggested, which have all been added into the new version of the manuscript.

Furthermore, the text of the paper is difficult to read. Sentences are fusions of several statements, figures are described in an unusual order (e.g. Figure 3E before Figure 3A), manipulations are not sufficiently explained (e.g. sna and twi RNA, not mentioned what the abbreviations mean, not described in the Materials and methods section), the Introduction is extremely brief consisting of 3 sentences followed by only 29 references. Also, the manuscript requires major editing.

We also have followed all reviewers' advice regarding editing. The long sentences that contained several statements were shortened and cut into distinct sentences. The inverted Figure numbers were reordered (except for Figure 4A, C which is before Figure 4B, C, because Figure 4C consists in the quantitative results of Figure 4A and Figure 4B merged in the same graph). Manipulations were further explained when required (e.g. for *twi sna* siRNA injections). And the Introduction was lengthened to introduce the context of the present finding in more detail.

Essential revisions:1) The lifetime differences of the antibody injected embryos are surprisingly small.

Indeed, the lifetime differences of the antibody-injected embryos are small (~0.1 ns), but significant (the error bars are on the order of ~0.05 ns and associated p-values are now of 0.004 for the mesoderm at stage 6 compared to stage 5). This is in fact in agreement with the molecular dynamics simulation, in which the strain induces small conformation changes on the order of 1 nm for the distance between the GFP and A555 two chromophores attached to the a-helixes connected by the β-cat Y654/E-Cad D665 link, at the opening of the β-cat Y654/E-Cad D665 site (Figure 1C). Such 1nm fine stretching leads to a life-time difference on the order of 0.1ns. This is given by the FRET efficiency equation r=R0((1/E)-1)^1/6 that links the distances r between the chromophores, and the donor lifetime tda (where E=1-tda/td, td being the lifetime of the donor alone and R0 is the Förster Radius, in the approximation that there are no cross-FRET interactions between adjacent molecules (Wallrabe and Periasamy, 2005). The Förster Radius for this particular FRET pair (GFP and Alexa 555) is R0=6.3nm (Thermofisher datasheet) (see **“**Calculation of the strain from FRET efficiency variations” in the Materials and methods section).

This relationship is now better explained in the Results section, in which the reference to Figure 1C, as well as to the Materials and methods section for calculations, were effectively both missing in the previous version of the manuscript:

“At stage 6, measurements showed an increase of the GFP lifetime in the mesoderm of 0.073 +/- 0.029 ns compared to the ectoderm, and of 0.094 +/- 0.059 ns compared to stage 5 mesoderm (2.473 ns) (Figure 2C). […] The lifetime increase observed specifically in the mesoderm at gastrulation thus shows the existence of a mechanical stretching strain between the fluorescently labelled linear domain II of E-cad, and the C-terminus of β-cat, respectively, in the invaginating mesoderm.”

In addition, given these small but significant differences, we increased the number of experiments (N=11), as suggested by the review below, to further test the robustness of the results. The results remained robust, with an increase of the statistical significance from p=0.0120 to 0.004 for the lifetime difference of the mesoderm at stage 6 compared to stage 5 (the p-value of the mesoderm versus the ectoderm at stage 6 remained on the same order of magnitude, of p=0.022 compared p=0.016 in the previous version of the manuscript). This is newly added in Figure 2C legend:

“c Left – Quantification of the lifetime difference at different stages and different tissues. […] No significant change was observed in the ectoderm at stage 6 (N=10) compared to stage 5 (N=8) (p=0.74), and between the mesoderm and the ectoderm at stage 5 (p=0.33). In some embryos, the ectoderm could not be imaged properly."

As it is very difficult to control the amount of antibody (and thus FRET acceptor) being present at different locations in the syncytial blastoderm one would expect much larger variations.

The reviewers are correct that fluctuations could be expected to be large, due to putative inhomogeneities in the labelling of the acceptor at junctions into the mesoderm. Indeed, fluctuations are relatively large, on the order 0.5 ns compared to the 0.1 ns of the mean values in Figure 2C and 3D.

However, we have verified that all junctions are labelled with the acceptor antibody (see Figure 2—figure supplement 1B). This observation indicates that fluctuations in the labelling, with variations in the FRET lifetime, should be smaller than under conditions including a mixture of non-labelled and labelled junctions. This possible origin of the observed fluctuations is now implemented in the Results section:

“Furthermore, despite the fact that all junctions are labelled with the E-cad Alexa 555 acceptor (see Figure 2—figure supplement 1B), modulations in the labelling efficiency may be at the origin of variations observed in the lifetime.”

To ensure that the reported values are reproducible the number of injected embryos should be increased (from N=8 embryos).

In fact, the number of embryos tested for single embryonic mechanical manipulation is generally of 5-8 for each condition in mechanobiology, because of the time consuming nature of the experiments (Figure 3 of ref^1^, Figure 4 of ref^2^ and Figure 4D of ref^3^). In the first version of the manuscript, we had generated 8 embryos for each condition. However, despite the significant p values of the results with 8 embryos, we agree that the fluctuations in the results discussed above suggested the conduction of additional experiments. We now have a total of N=11 embryos in which the lifetime increase was measured at stage 6 in the mesoderm during its invagination.

As mentioned above, the results remained robust, with an increase of the statistical significance from p=0.0120 to 0.004 for the lifetime difference of the mesoderm at stage 6 compared to stage 5. The p-value of the mesoderm versus the ectoderm at stage 6 remained on the same order of magnitude, of p=0.021 compared to p=0.016 in the previous version of the manuscript. This information is now added in the Figure 2C and associated legend:

“C Left – Quantification of the lifetime difference at different stages and different tissues. […] As expected, PBS injection without the antibody acceptor lead to a lifetime of 2.6 +/- 0.039 ns, higher compared to 2.473 +/- 0.074 ns with the antibody at stage 5 (see Figure 2—source data 1, p=0.00016). Statistical tests are t-student.”

Furthermore, an injection control is necessary, as uninjected embryos are not sufficient to control the experiment. Does water/PBS injection have an effect? Does this vary between the stages?

We agree with the review comment. We have thus conducted new experiments in which embryos were injected with PBS alone. As expected, PBS injection without the antibody lead to a lifetime of 2.6 +/- 0.039 ns, larger compared to 2.47 +/- 0.074 ns with the antibody at stage 5 ((p=0.00016) see Figure 2—source data 1). In addition, PBS injected embryos did not show any variation of the lifetime during gastrulation stage 6, compared to before at stage 5 (see Figure 2C-right beyond). Together with the non-injected β-cat-GFP control embryos, these new controls further confirm the presence of the acceptor labelled antibody as required for the observed increase in GFP life-time in the mesoderm during its invagination compared to before. This new control is now shown as new Figure 2C-right, and described as follows:

“As a control, no change in the GFP donor lifetime was observed during gastrulation in the absence of injected antibody acceptor (E-cad Alexa 555 labelled AB), as well as in embryos injected with PBS alone in the absence of the antibody acceptor (Figure 2—figure supplement 2A, B and Figure 2C-right).”

and in the Figure 2C legend:

“As expected, PBS injection without the antibody acceptor lead to a lifetime of 2.6 +/- 0.039 ns, higher compared to 2.473 +/- 0.074 ns with the antibody at stage 5 (see source data, p=0.00016). Statistical tests are t-student.”

How was the lifetime fit done? Is it a mono-exponential fit?

The lifetime was fitted with a mono-exponential. This information, that was effectively missing, is now added in the related Materials and methods section.

Is it not surprising that the GFP donor lifetime is <2.6 ns in the absence of an acceptor? This is impressively short!

In the absence of acceptor, we measured a GFP lifetime of 2.6 +/- 0.04 ns at any stage (Figure 2—figure supplement 2B and Figure 2—source data 1), which is consistently higher than the lifetime measured at stage 5 before the mechanical strains of mesoderm invagination, of 2.473 +/- 0.074 ns (Figure 2C and Figure 2—source data 1). Moreover, these values are on the order of magnitude of the lifetime of GFP tagged proteins measured in cellula, for instance of 2.6 ns coupled to Histone in the nucleus in the absence of an acceptor, and of 2.4 ns in the presence of an acceptor (Lleres et al., 2017).

This information has been added:

“Note that the lifetime of the controls with GFP alone in the embryo junctions and with PBS injection without antibody, of 2.60 +/- 0.04 ns, (Figure 2—figure supplement 2A, B and Figure 2—source data 1C-right), was consistently higher than in the presence of the acceptor before stretching, of 2.473 +/- 0.074 ns (Figure 2—source data 1C-left). These values are similar to the lifetime of GFP tagged proteins measured in cellula, namely of 2.6ns coupled to Histones in the nucleus in the absence of acceptor and of 2.4ns in the presence of an acceptor (Llleres et al., 2017).”

2) Why are double RNAi against twi and sna injected instead of using zygotic double mutants?

Zygotic double mutants were not used because existing *twi sna* zygotic mutants do not include the Arm-GFP (β-cat-GFP). To generate the cross between the *sna twi* /Cyo double zygotic mutants and Arm-GFP flies would have significantly delayed the project. Indeed, the *twi sna* RNAi treatment has already been described as efficient in blocking gastrulation just like *twi sna* double zygotic mutants (Mitrossilis et al., 2017, Martin et al., 2009). Furthermore, injecting siRNA for *twi* and *sna* to induce *sna twi* RNAi in Arm-GFP embryos leads directly to *sna twi* gastrulation defective embryos in the presence of Arm-GFP.

This information is now newly detailed with the RNAi sequences:

“RNAi

To produce *twi* and *sna* defective non gastrulating Arm-GFP embryos, we injected siRNAs for *twi* and *sna* RNAi into Arm-GFP embryos. *twi sna* RNAi embryos have been described to phenocopy *twi sna* double zygotic mutants, and do not gastrulate^6,7^. 3 siRNA sequences were designed and produced by Eurogentec, and already used in Mitrossilis et al, 2017:

*sna*-siRNAs (1481-1503 S 5': CAGCUAUAGUAUUCUAAGU dTdT; AS 3': ACUUAGAAUACUAUAGCUG dTdT, 340-362 S 5': GGAACCGAAACGUGACUAU dTdT; AS 3': AUAGUCACGUUUCGGUUCC dTdT, 972-994 S 5': CCGAGUGUAAUCAGGAGAA dTdT; AS 3': UUCUCCUGAUUACACUCGG dTdT)

*twi*-siRNAs (928-950 S 5': GCACCAGCAACAGAUCUAU dTdT; AS 3': AUAGAUCUGUUGCUGGUGC dTdT, 2061-2083 S 5': GUCACGCUUUCCAUAUAUA dTdT; AS 3': UAUAUAUGGAAAGCGUGAC dTdT; 1-1993 S 5': CGGAUCAGGACACUAUAGU dTdT; AS 3': ACUAUAGUGUCCUGAUCCG dTdT)”

As well as, following, the conditions of injection with the UMLs described in Mitrossilis et al, 2017, and the characteristics of the magnetic conditions used to produce the deformation that mimics the initiation of apex constrictions initiating mesoderm invagination.

The small differences the authors observe upon RNAi injections could be due to differential knock-down efficiencies, particularly if the number of treated embryos is so low (N=8).

There was effectively an inhibition of the lifetime increase observed at stage 6 compared to stage 5 embryos, in the *twi sna* RNAi compared to the WT embryos (Figure 3E to be compared to Figure 2C, of the previous version of the manuscript). Despite the small number of embryos of the *twi sna* RNAi condition of the previous version of the manuscript (N=4), this in fact already showed a full inhibition of the increase of the lifetime in the *twi sna* RNAi injected embryos compared to the WT embryos of Figure 2c, at stage 6 compared to stage 5, with a statistically relevant difference between the two conditions at stage 6 with a p-value of p=0.016.

This p-value was effectively not provided in the previous version of the manuscript. It is now present in the new version of the manuscript (Figure 3D of the present version of the manuscript legend).

This p value is now upgraded to p=0.0003 by following the reviewer’s advice to increase the number of experiments (now N=11 in the WT and N=8 in the *twi sna* RNAi without magnet, and N=9 in the *twi sna* RNAi in the presence of the magnet), which were effectively relatively low in the previous version of the manuscript. The legend of Figure 3D thus now states:

**“**d Lifetime difference of stage 6 to the average life-time before magnetic field application of stage 5, of N=9 experiments with magnet applied and N=8 without magnet applied as well as N=6 experiments with PBS instead of antibody injected and magnet applied. […] The absence of increase of the life-time difference within *sna twi* RNAi non-invaginating conditions without magnet between stage 5 and stage 6 is characterized by the p-value p=0.0003 compared to the increase between stage 5 and 6 in the invaginating WT of Figure 2C-left, and of p=0.75 compared to the lifetime difference in the invaginating WT injected with PBS of Figure 2C-right (t-students test).”

3) The rescue with the magnet should also be controlled with water injected embryos.

We agree that this control was missing. The magnet experiment was thus performed in the absence of the antibody, with PBS and UML injection alone. No increase or modification in the lifetime was observed (Figure 3D). This observation confirmed the requirement of the acceptor to observe an increase in the lifetime in the mesoderm under magnetically controlled forces mimicking the apex constriction at the onset of gastrulation in *twi sna* RNAi non-gastrulating embryos. It is like the requirement of the acceptor to observe an increase in the lifetime in the mesoderm under endogenous gastrulation constraints in the WT, at stage 6 (Figure 2—figure supplement 2B and Figure 2C)). This information was added to Figure 3D:

“d Lifetime difference of stage 6 to the average life-time before magnetic field application of stage 5, of N=9 experiments with magnet applied and N=8 without magnet applied as well as N=6 experiments with PBS instead of antibody injected and magnet applied. […] The absence of increase of the life-time difference within *sna twi* RNAi non-invaginating conditions without magnet between stage 5 and stage 6 is characterized by the p-value p=0.0003 compared to the increase between stage 5 and 6 in the invaginating WT of Figure 2C-left, and of p=0.75 compared to the lifetime difference in the invaginating WT injected with PBS of Figure 2C-right (t-students test).”

And in the Results:

“As controls, no increase in the lifetime between stage 5 and stage 6 was observed after injection of the UMLs and RNAi without magnet (Figure 3D), or with magnets after UML and PBS injection without the E-cadh Alexa 555 acceptor (Figure 3D).”

4) Quantification of the Y654 antibody stainings also requires a control experiment. The authors observe an increase in Y654 non-phospho antibody staining during gastrulation or upon Src knock-down. The authors seem to have normalized the Y654 non-phospho staining to general β-catenin antibody staining. An additional control, just outside of the mesoderm from the same embryo should be included, as the forces on the lateral cells should be less, the difference should also be less. Since the antibody signals can vary between embryos, such an internal standard makes it easier to compare different embryos.

We agree that an internal control measurement in the ectoderm should re-enforce the significance of the results. We have thus measured Y654 non-phospho relative to general β-cat into the ectoderm at stage 6 compared to stage 5. Indeed, consistent with the force being less important in the ectoderm than in the mesoderm at gastrulation, we found no significant increase in both the WT and Src42A knock-down conditions (new Figure 4D). We additionally normalized the mesoderm Y654 antibody staining with the ectoderm Y654 antibody staining outside of the mesoderm. This procedure consistently showed an increase of the signal in both the WT and Src42A knock-down conditions (new Figure 4E). This, with associated p-values, is now implemented in Figure 4:

“Figure 4: Increase of accessibility of Y654-β-cat in the mesoderm during invagination. […] WT compared to *src42A* RNAi early stage 6 p=0.049, late stage 6 p=0.063, difference between the two curves p=0.021.”

5) Is it possible to design a control experiment with mutant, stretch-resistant DE-cadherin and/or β-catenin and show that if the sites are made resistant to stretching, accessibility of the Y654 site is unaffected by the application of strain?

We thank the reviewers for this suggestion. Its implementation, however, would have been excessively difficult to achieve in 3 months. Based on such a suggestion, we would propose, for future investigations, the design of relatively soft E-cad/β-cat proteins, for instance by adding soft coiled elements in the most linear domain of the E-cadherin between A648 and the membrane. All of the mechanical work energy being first of all stored in the coiled protein domain conformation change, this design would prevent the force to apply efficiently at the Y654-D655 site and to open it.

6) The JCAb20 antibody recognises numerous bands in addition to the DE-cadherin (peptide). Based on the unspecific bindings it is not clear how valid the findings with this antibody are?

We agree that the multiple non-specific bands of the western blot may suggest the possibility that the findings might have been independent on the specific labelling of E-cadherins. However, immunofluorescence labelling clearly shows a labelling specific of junctions, fully co-localized with β-catenin, and measurements were realised focused on the junctions plan only.

To fully exclude the labelling of another junctional protein, that would only have been stabilized and observed if it was concentrated close enough to the β-catenin-GFP fluorophore (within several nm only) to show a FRET effect, we performed an immunoprecipitation assay with the JCAb20 antibody acting as trap by being attached to protein-A agarose beads. We found a single band in western blot recognized by the DE-cadherin antibody at the 140kDa molecular weight, which corresponds to the full length DE-cadherin protein, as newly introduced in Figure 2—figure supplement 1A (additional bands around 40 kDa are presumably the immunoglobulin heavy chains with different glycosylations). Note that we could properly synchronize 70 embryos at stage 5 for the immunoprecipitation experiments, which represents a relatively low amount of biological materials, and explains the relatively slight band of immunoprecipitated material.

In contrast to the western blot procedure, immuno-precipitation uses non-de-naturating procedures for proteins, therefore the additional bands observed in the western blot are non-specifically associated to denatured proteins. This new result definitively demonstrates the high specificity of the DE-cadherin labelling by the JCAb20 antibody. It is now added to Figure 2—figure supplement 1A:

“Figure 2—figure supplement 1: Specific labeling of junctional E-cad with JCAb20. […] Right: Immunoprecipitation by using the JCAb20 antibody as a trap, and the DE-cad antibody for revelation. The band is at the 140kDA weight of the mature DE-cad protein. Other bands around 40kDA are presumably immunoglobulins. Colored bands are pre-stained markers.”

as well as newly described as follows:

“The antibody was tested for its specificity by immuno-staining, western-blot analysis and immunoprecipitation (see Figure 2—figure supplement 1A, B). The peptide sequence of the immunogen was checked for possible similarities with other proteins by a standard protein-protein BLAST search against the *Drosophila* proteome (taxid: 7227) (see Figure 2—figure supplement 1C).”

with the experimental procedure described in the Materials and methods section.

7) Do the authors have an explanation why 85% of β-catenin does not alter the association with DE-cadherin during mechanotransduction? How do these 85% of β-catenin escape the dissociation and phosphorylation by Src?

Our interpretation is that, once under tension, the deformation of the E-cadherin/β-cat complex can a priori statistically occur at different locations all along the complex structure, including at the Y654-D665 site. We further assume that in 15% of cases, the deformation occurs at the Y654-D655 site and opens it. Future investigations on the simulations would be required to quantitatively test this hypothesis.

This interpretation is now added to the Discussion:

“A possible interpretation for the observed 15% efficiency of site opening is that, once under tension, the deformation of the β-cat/E-cad complex can a priori statistically occur at different locations all along the complex structure. […] Future molecular dynamics investigations will be required to quantitatively test this hypothesis.”

1) Hutson, M. S. et al. Forces for morphogenesis investigated with laser microsurgery and quantitative modeling. Science 300, 145-149 (2003).

2) Fernandez-Gonzalez, R., Simoes Sde, M., Roper, J. C., Eaton, S. and Zallen, J. A. Myosin II dynamics are regulated by tension in intercalating cells. Dev Cell 17, 736-743 (2009).

3) Yu, J. C. and Fernandez-Gonzalez, R. Local mechanical forces promote polarized junctional assembly and axis elongation in Drosophila. *eLife* 5, doi:10.7554/eLife.10757 (2016).